# Dimensions of wisdom perception across twelve countries on five continents

M. Rudnev [1] ✉, H. C. Barrett [2], W. Buckwalter[3], E. Machery[4,5], S. Stich[6], K. Barr [4], A. Bencherifa[5,7], R. F. Clancy[8], D. L. Crone[9], Y. Deguchi[10], E. Fabiano [11,12], A. D. Fodeman[13], B. Guennoun [14], J. Halamová [15], T. Hashimoto[16], J. Homan[17], M. Kanovský[15], K. Karasawa[18], H. Kim [19], J. Kiper [20], M. Lee [21], X. Liu[22], V. Mitova [5], R. B. Nair[5,23], L. Pantovic[24], B. Porter [4], P. Quintanilla [5,12], J. Reijer [5], P. P. Romero [25], P. Singh[23], S. Tber [7], D. A. Wilkenfeld[4], L. Yi[4] & I. Grossmann [1,5] ✉

Wisdom is the hallmark of social judgment, but how people across cultures recognize wisdom remains unclear—distinct philosophical traditions suggest different views of wisdom's cardinal features. We explore perception of wise minds across 16 socio-economically and culturally diverse convenience samples from 12 countries. Participants assessed wisdom exemplars, non-exemplars, and themselves on 19 socio-cognitive characteristics, subsequently rating targets' wisdom, knowledge, and understanding. Analyses reveal two positively related dimensions—Reflective Orientation and Socio-Emotional Awareness. These dimensions are consistent across the studied cultural regions and interact when informing wisdom ratings: wisest targets—as perceived by participants—score high on both dimensions, whereas the least wise are not reflective but moderately socio-emotional. Additionally, individuals view themselves as less reflective but more socio-emotionally aware than most wisdom exemplars. Our findings expand folk psychology and social judgment research beyond the Global North, showing how individuals perceive desirable cognitive and socio-emotional qualities, and contribute to an understanding of mind perception.

Philosophers from various cultural traditions have proposed a range of features central to wisdom, from critical thinking and self-awareness to spirituality and social intelligence. Differences in philosophical traditions across cultures suggest that lay understanding of who is wise— i.e., the social perception of wisdom—may vary greatly between

societies, as well. Just like philosophical traditions vary in their emphasis on rational deliberation or social context sensitivity for good judgment[1–3], some cultures, particularly those emphasizing individualist values, tend to prioritize analytic and reflective skills in their perception of wisdom, often regarding emotions and sensitivity to

[1]University of Waterloo, Waterloo, ON, Canada. [2]UCLA, Los Angeles, CA, USA. [3]George Mason University, Fairfax, VA, USA. [4]University of Pittsburgh, Pittsburgh, PA, USA. [5]University of Johannesburg, Johannesburg, South Africa. [6]Rutgers University, New Brunswick, NJ, USA. [7]Université Internationale de Rabat, Rabat, Morocco. [8]Virginia Tech, Blacksburg, VA, USA. [9]Northeastern University, Boston, MA, USA. [10]Kyoto University, Kyoto, Japan. [11]University of Coimbra, Coimbra, Portugal. [12]Pontificia Universidad Catolica del Peru, San Miguel, Peru. [13]Centers for Disease Control and Prevention (CDC), Atlanta, GA, USA. [14]Ibn Tofail University, Kenitra, Morocco. [15]Comenius University in Bratislava, Bratislava, Slovakia. [16]Toyo University, Tokyo, Japan. [17]University of Kansas, Lawrence, KS, USA. [18]The University of Tokyo, Tokyo, Japan. [19]Korea University, Seoul, South Korea. [20]University of Alabama at Birmingham, Birmingham, AL, USA. [21]Seoul National University, Seoul, South Korea. [22]Wuhan University, Wuhan, China. [23]Indian Institute of Technology, Delhi, India. [24]University of Belgrade, Belgrade, Serbia. [25]Universidad San Francisco de Quito, Quito, Ecuador. ✉e-mail: maksim.rudnev@uwaterloo.ca; igrossma@uwaterloo.ca

contextual factors as less relevant[4]. In contrast, cultures that emphasize collectivist values may inherently value socio-emotional competencies, even when these competencies may appear to contravene principles of logic and rational deliberation[5]. From this perspective, one can expect sizable and systematic cultural differences in wisdom perception along cultural dimensions of individualism or collectivism[6]. Moreover, anthropologists and cultural psychologists have observed that people from different cultures pay varying degrees of attention to emotions, thoughts, and bodily sensations and may even lack specific terminology to describe certain mental states[7–10]. In some cases, what one society considers wise could even be viewed as foolish by another[11]. Indeed, the very notion of 'wisdom' might be a culturally specific construct, lacking an exact equivalent in some societies.

Contrary to such highly plausible cultural differences in wisdom perception, some theories in cognitive, developmental, and social psychology hint at a possible convergence in judgment of wisdom across cultures. First, wisdom perception is a form of social judgment that concerns desirable mental states[6]. A range of cognitive and developmental psychologists have proposed that reflection on mental states of others and oneself is a central part of most human lives and hence may be universal[12–14], with empirical scholarship suggesting two dimensions along which people evaluate perception of others' minds[15]: intentional agency (the capacity to engage in reasoned action, self-control, strategic planning, or goal-directed behavior) and conscious experience (metacognitive capacities, including secondary emotions, conscious awareness of one's environment, and basic psychological states). Supporting these claims, empirical scholarship has identified cross-cultural similarity in the perception of cognitive abilities; in particular, reasoned action appears distinct from experiences such as bodily sensations across five societies[16]. Second, people often inflate their competencies on characteristics they view as central to their self[17–19]. To the extent that people from different cultures consider wisdom-related characteristics as desirable[20–23], social judgments of others versus the self could be influenced by self-enhancement processes similar to those identified in prior research. Third, theorists of social judgment have asserted that people in most cultures use similar dimensions of person perception, assessing people based on their ability to master tasks and their ability to coordinate with others[24]. Thus, the underlying general dimensions of social judgment—analytical competencies and socio-emotional experiences—could influence wisdom perception similarly across cultures.

However, the empirical support for either cultural diversity or cross-cultural convergence in dimensions of wisdom perception remains inconclusive. First, most of the existing research on social judgment comes from European and North American countries, with a dearth of research on social judgment in the Global South. Only one empirical study attempted to evaluate dimensions of social judgment beyond WEIRD societies by including three East Asian samples[25], yet it suffered from insufficient statistical power (60-91 participants per site), methodological limitations (e.g., no formal testing of the measurement model, presence of response bias/halo effect), and poor representation of the Global South.

Second, whereas initial research on perception of mental states has identified two dimensions described as intentional agency (encompassing reasoned action and cognitive control) and conscious experiences (encompassing meta-cognition and awareness of one's body and environment[15]), subsequent work has identified one dimension (no mind vs. mind[26]), two dimensions[27], or three dimensions of mind, corresponding respectively to physiological abilities, socio-emotional abilities, and cognitive abilities[28]. Again, most of these studies have been limited to WEIRD samples. When a non-WEIRD comparison was included, a more nuanced pattern emerged (e.g., a separate dimension of social relationships emerged in Fiji[27]). As we saw, one notable exception involved a recent five-country study across the US, Ghana, Thailand, China, and Vanuatu[16], revealing some

consistency in differentiation in the perception of mind- and body-related capacities, but cultural differences in the relevance of the heart-related (socio-emotional) capacities. However, this study relied on a simplified verbal protocol and did not adequately test the cross-cultural invariance of their model—a necessary condition for comparison.

Third, there are good reasons to expect that social judgment about group stereotypes or about broad mental states of humans, robots, and non-human animals do not reflect processes guiding the evaluation of psychological characteristics in the situations typically calling for wisdom (e.g., the context of decision-making under uncertainty). The dimensions guiding abstract trait ascription to stereotyped groups (e.g., warm, trusting, selfish, and cold) in prior social judgment research may be psychologically distinct from the dimensions used to evaluate individuals' concrete mental states implicated in wise judgment under uncertainty[6]. Furthermore, the perception of wise minds may be distinct from the perception of a typical human mind, similar to the differences in perceptions of a divine and a typical human mind[27]. Critically, when squarely focusing on surveys of folk theories of wisdom, some previous research suggests that Western cultures emphasize cognitive characteristics in their definitions of wisdom more than non-Western cultures[4], whereas other scholarship shows that Western cultures also emphasize socio-emotional characteristics[29].

Finally, even if dimensions informing wisdom perception were to be cross-culturally invariant, their application in evaluating others' wisdom may vary across cultures. For instance, some cultures may use these dimensions in a synergistic fashion—each dimension contributes to the perception of wisdom only if one is also perceived as high on the other dimension. Other cultures may view one dimension (e.g., 'rational mind') as a necessary source in evaluating one's wisdom, treating another dimension (s) (e.g., 'socio-emotional experience') as a secondary indicator; all other things being equal, the second indicator may even detract from the perception of wisdom, as is the case when considering folly in decisions hyper-focused on emotional responses to specific individuals rather than reasoned considerations of overall impact or broader ethical principles[30]. To our knowledge, no existing scholarship has so far systematically evaluated the application of dimensions governing social or mind perception across cultures.

Bringing these strings of scholarship together, we sought to build on a bottom-up approach used in prior scholarship on perception of mental states[15,28], simultaneously addressing a range of methodological and cross-cultural limitations identified in prior scholarship. To this end, we evaluated the perception of wisdom in others and the self across 16 samples from 8 distinct cultural regions.

Our research aimed to investigate the latent dimensions that guide people's evaluation of characteristics associated with wisdom and whether these dimensions are consistent across cultures. We also examined the relationship between these latent dimensions and the explicit attribution of wisdom and closely related epistemic attributes (knowledge and understanding) to specific individuals and oneself. To accomplish these aims, we developed an instrument that prompted participants to compare ten individuals, including themselves, in the context of making a difficult choice without a clear right or wrong answer. First, participants were provided with concrete descriptions of specific individual targets (e.g., 'Dr. [name] is a scientist who gathers information about plants, animals, and people to make sense of the world'). Next, they compared pairs of such targets (e.g., a scientist to a teacher) on 19 characteristics associated with wisdom in prior philosophical and psychological scholarship on wisdom (e.g., 'think logically,' 'pay attention to others' perspectives'). We took care to include characteristics capturing wisdom-related mental states corresponding to mind-, heart-, and body-related themes identified in prior research[15,16,27,28]. This approach resulted in up to 171 pairwise comparisons (see Fig. 1 for study flow). The instrument also allowed us to

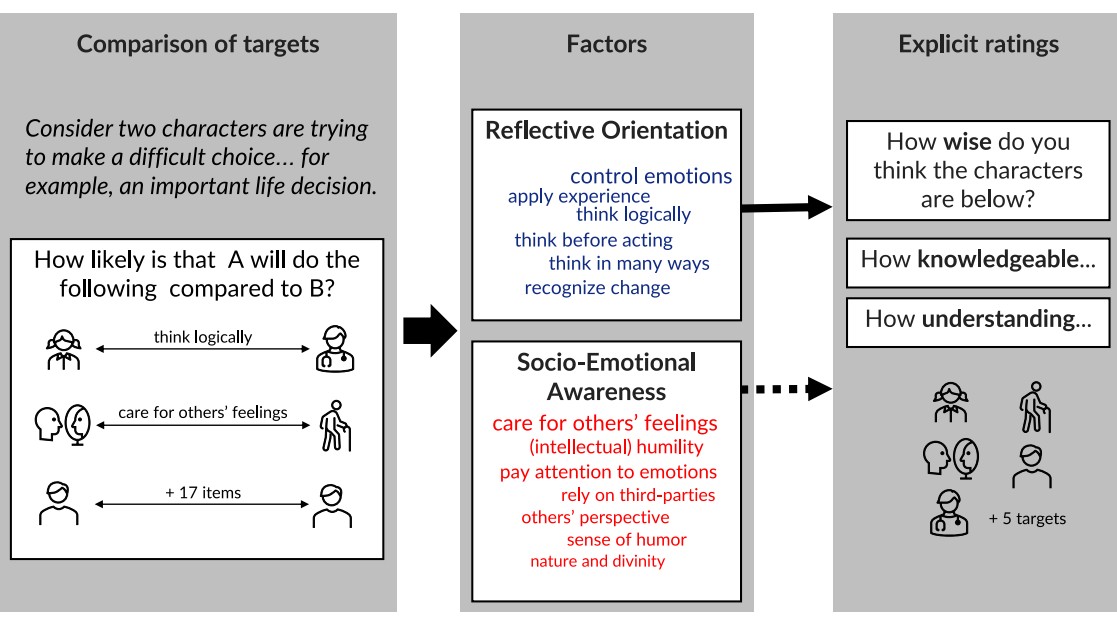

**Fig. 1 | Study flow.**

examine cultural differences while accounting for response bias and other possible between-person differences across sites (e.g., gender, age, and education). Finally, we asked participants to explicitly rate targets' wisdom and related epistemic characteristics of knowledge and understanding. We gathered data from 16 samples across 11 languages and five continents.

Probing a range of models proposed in prior research on mind perception[15,26,28] and social judgment[24,25,31], we observed two latent dimensions that guide people's evaluation of wisdom-related characteristics in others and the self—reflective orientation and socio-emotional awareness, which were aligned with the explicit attribution of wisdom (as well as knowledgeability and understanding) to specific individuals. Contrary to our expectations, these dimensions were consistent across cultures. Moreover, ratings of targets were stable across cultures on the Reflective dimension but varied depending on culture on the socio-emotional dimension. Additionally, we found that people in most cultures compared themselves favorably on socio-emotional characteristics associated with wisdom vis-à-vis exemplars of wisdom.

## Results

### Two dimensions of wisdom perception

Based on prior cross-cultural research[32,33], we grouped our samples into eight cultural regions. Participants compared ten human targets (including themselves) in a pairwise manner: They were asked whether one would be more likely than the other to act in a certain way when facing a difficult choice where there is no clear answer (e.g., 'think logically,' 'care for others' feelings;' see Fig. 1 and Methods for further details). Each of the nineteen actions reflected a wisdom-related characteristic as discussed in prior research[21,34].

The target comparisons formed a multi-level dataset, with ratings of different targets by each characteristic nested within participants. We submitted this data to a series of factor analyses (Fig. 1). Model fit was evaluated with Comparative Fit Index (CFI, values greater than 0.9 signal an acceptable fit[35]), Root Mean Square Error of Approximation (RMSEA < 0.08), and Standardized Root Mean Error (SRMR < 0.08, reported for each level separately). In the first step, we aimed to identify the most stable configuration of factors—i.e., factors that remained consistent across exploratory multilevel factor models with a different number of factors and provided interpretable solutions (as reflected in a meaningful combination of

items) in each cultural region. This iterative process revealed an acceptable two-factor solution (see Supplementary Note A). By virtue of pairwise comparisons, our method controlled for an acquiescent response style within individuals. At the between-participant level, the presence of response tendencies was tested by the introduction of a method factor which improved the model only negligibly (difference ($\Delta$) in CFI, RMSEA, and $SRMR_{within}$ was less than 0.001; and $\Delta SRMR_{between} = 0.006$). Striving for a more parsimonious model, we thus omitted the method factor from further analyses (see methods for further details).

In the second step, we explored whether the two-factor model would be best described by factor loadings that are isomorphic across within- and between-individual levels of analyses. Isomorphic models assume equal loading across levels of analysis and are, therefore, more parsimonious[36]. Cross-level isomorphism implies that the psychological processes underlying the attribution of characteristics to targets within an individual are similar to those defining between-individual differences. Cross-level isomorphism also implies that the constructs measured within and between individuals are comparable. Comparison of an isomorphic model constraining factor loadings across levels of analyses and a non-isomorphic model allowing them to vary demonstrated similarly good fit to the data, CFI = 0.956 and 0.963; RMSEA = 0.022 and 0.021; $SRMR_{within} = 0.028$ and 0.024, $SRMR_{between} = 0.082$ and 0.036, respectively. Thus, we proceeded with a more parsimonious isomorphic model. Since the within-individual level dominated the model (80% of the variance in the data), as indicated by the intraclass correlation between 0.23 and 0.29 across different items, the deviations from isomorphism would be able to bias the parameters at the between level only. Importantly, the within-individual structure (including specific factor loadings) was unaffected by these model modifications. Thus, hereafter we focus on the results on the within-individual level (results on the between level are very similar; see SI).

To interpret the meaning of each factor, we examined factor loadings (Fig. 2). We labeled the first-factor Reflective Orientation, with high loadings of characteristics concerning thinking before acting, thinking logically and in many ways, recognition of change, emotion control, as well as application of knowledge and past experience. Overall, this dimension integrates pragmatic, rational, analytic, and self-control traits. This factor resembles some features discussed in prior research on mind perception

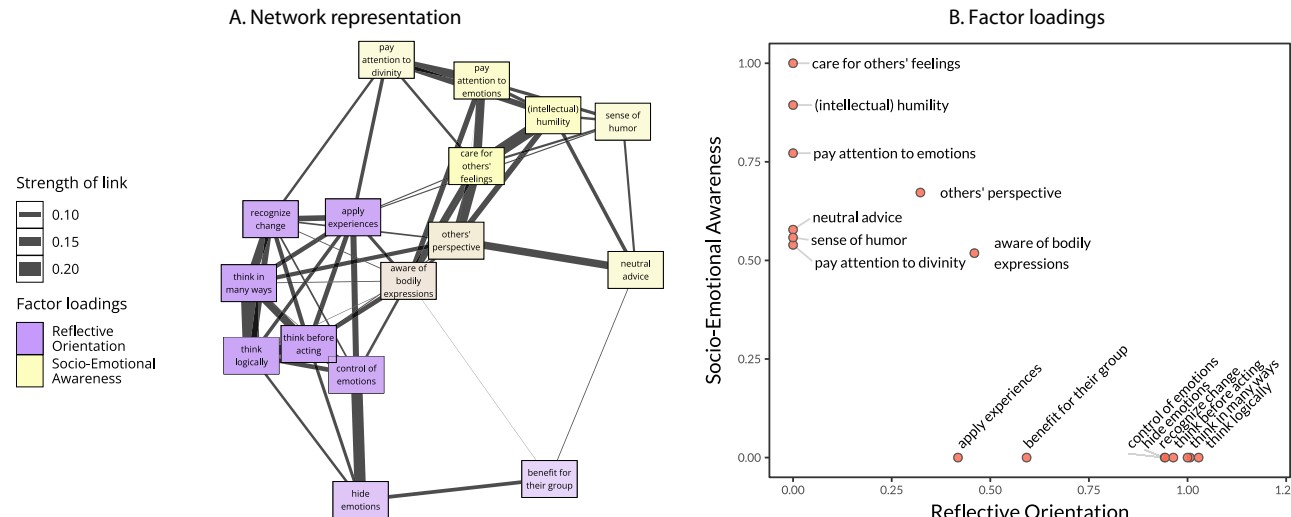

**Fig. 2 | The structure of the latent wisdom perception dimensions. A** Network graph representation of items demonstrating closer (and stronger) associations of items making up each factor. **B** Unstandardized factor loadings of items of the two factors taken from a multigroup multilevel confirmatory factor analysis. Drawing on prior tests, the underlying model assumes isomorphism (i.e., equal factor loadings at between- and within-individual levels) and partial invariance of loadings across eight cultural regions (only loadings on items 'aware of bodily expressions,' 'consider others' perspective,' and 'listen to nature or divinity' differed across cultural groups). The model fit was acceptable, CFI = 0.912, RMSEA = 0.033, SRMR$_{within}$ = 0.032, SRMR$_{between}$ = 0.078.

**Table 1 | Cross-cultural measurement invariance tests of the two-dimensional model of wisdom perception**

| | BIC | CFI | Δ | TLI | Δ | RMSEA | Δ | SRMR$_{within}$ | SRMR$_{between}$ |
|---|---|---|---|---|---|---|---|---|---|
| Configural | 1,079,658 | 0.922 | | 0.918 | | 0.032 | | 0.026 | 0.071 |
| Partial metric[a] | 1,080,415 | 0.912 | 0.010 | 0.910 | 0.008 | 0.033 | 0.001 | 0.032 | 0.078 |
| Full metric | 1,081,374 | 0.901 | 0.021 | 0.901 | 0.017 | 0.035 | 0.003 | 0.034 | 0.081 |

*Note*: Tests of the isomorphic model without the method factor.
[a]Loadings of three items were estimated freely across regions: (1) 'consider someone else's perspective;' (2) 'pay attention to what nature or divinity is telling them;' and (3) 'be aware of bodily expressions.'

and social judgment, which suggested 'intentional agency'/'mind' (or 'competence') as one dimension of the judgment of mental states and groups.

We labeled the second factor Socio-Emotional Awareness because of the highest loadings of characteristics concerning care for others' feelings, one's emotions, and others' perspectives, as well as humility (recognition that one might be wrong). This factor appears similar to the 'conscious experience' that encompasses features of social metacognition, attention to the context, and one's emotions, as identified in some prior mind perception research[15]. Notably, it combined both features of 'heart'/socio-emotional characteristics and the 'body'-related experiences[27,28]. Taken together, these characteristics describe traits concerned with social coordination and care for others. On a broader conceptual level, these social-cognitive competencies are also related to the dimension of 'communion' identified in group-based social judgment research[24].

**Probing cross-cultural differences**

How stable is the two-dimensional model of wisdom perception across cultures? To address this question, we tested the invariance of the two-factor model across cultural regions. Results in Table 1 demonstrate a partial metric invariance, ΔCFI = 0.010; ΔRMSEA = 0.008; ΔSRMR$_{within}$ = 0.006; ΔSRMR$_{between}$ = 0.007. It implies that factor loadings were similar across the eight cultural regions. However, factor loadings of 'paying attention to nature and divinity,' 'consideration of others' perspective,' and 'awareness of bodily expressions' showed some variability across regions (Fig. S2 in the SI). Though speculative, cross-site variability in the value of 'nature and divinity' for the Socio-

Emotional Awareness dimension may reflect a stronger socio-cultural emphasis on nature and divinity in more traditional communities in Indian (Meitei) and South African (isiZulu and Sepedi) samples—the outliers in this item's loadings on the socio-emotional awareness dimension. We also observed some cultural variability in factor loadings of 'others' perspectives,' which did not form a meaningful pattern. Here, Chinese and Indian samples were outliers from other regions with diametrically opposite results: While the Chinese loadings were high and positive on the socio-emotional awareness and negative on the reflective orientation dimension, the Indian loadings were positive on the Reflective Orientation dimension, and around zero (and lowest compared to other groups) on the socio-emotional awareness dimension.

Though targets and characteristics varied widely, in most cultural regions perception of higher reflective orientation went hand in hand with higher perception of socio-emotional awareness, $r = 0.69$, 95% CI [0.66–0.71], $t = 63.0$, $p < 0.001$ (also see Fig. 5). This observation is consistent with classic work on person perception, suggesting a halo effect in evaluation of others (Rosenberg et al.[37]) due to an overall positive appraisal of exemplars. It may imply that participants focused on the holistic differences between targets rather than specific differences between characteristics they rated targets on (i.e., showing little discrimination between characteristics). This holistic association between dimensions was more pronounced in East Asian and South African regions, $0.76 < rs \leq 0.88$, compared to the Americas and North Africa, $0.33 < rs \leq 0.77$ (see also Table 2), in line with prior observation of cultural differences in holistic versus analytic perception between these cultural regions[38].

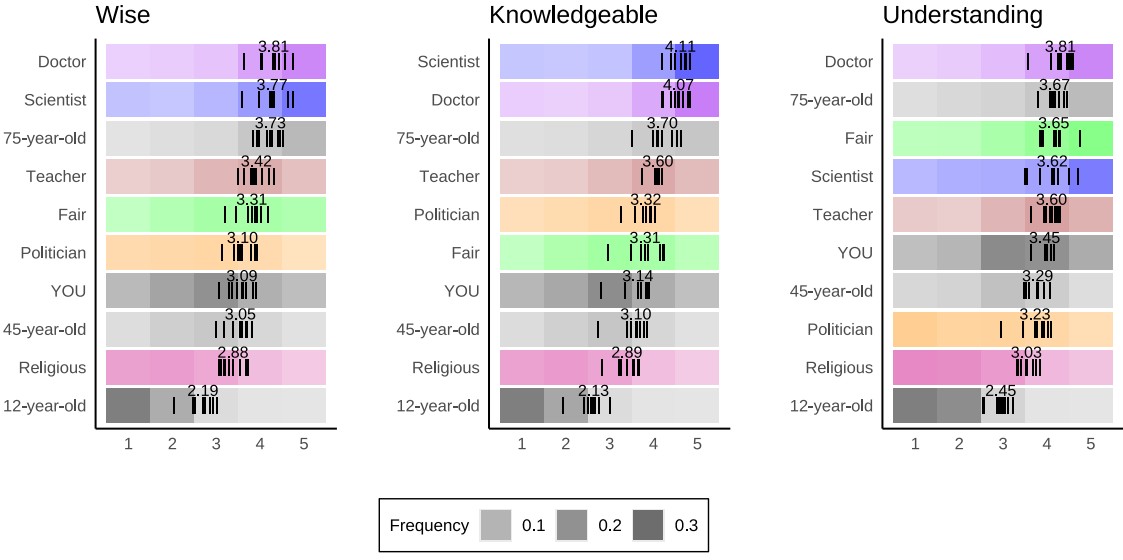

**Fig. 3 | Frequencies and means of wisdom, knowledgeability, and understanding for each target.** Vertical ticks represent mean ratings in each cultural region. Higher intensity of color represents higher frequency.

We performed several robustness checks of our results. First, we excluded the 12-year-old target, a possible outlier in the current set of targets. It somewhat decreased the factor model fit, $\Delta$CFI = 0.027, $\Delta$RMSEA = 0.003, while after the exclusion of 'self' ratings from the data, the model fit decreased only negligibly, $\Delta$CFI = 0.002, $\Delta$RMSEA = 0.002. Importantly, these analyses on restricted datasets barely changed the factor loadings of specific characteristics. The positive association between the two latent dimensions slightly decreased after the exclusion of age-specific targets and the self, yet still remained in a moderate-high effect size range, $0.63 < r$s $\leq 0.68$. Repeating these robustness checks within each cultural region showed similar results—i.e., we observed some decrease in model fit but virtually unchanged factor loadings (see results in Tables S20 and S21 in SI).

**Attribution of wisdom and related epistemic content**

Though the two latent dimensions of wisdom perception—Reflective Orientation and Socio-Emotional Awareness— appeared in each cultural region, we also whether these dimensions align with the explicit judgment of the targets' wisdom. Therefore, in the next step, we asked participants to rate each target's wisdom. To examine whether attributions of wisdom are idiosyncratic, we also asked participants to indicate how knowledgeable and understanding they perceived each target to be—that is, epistemic characteristics invoked in many cultures when mentioning wisdom[21,34] (see Supplementary Note D). The ordering of ratings of wisdom, knowledge, and understanding were randomized to avoid carry-over and contrast effects. Comparison of target ratings for wisdom, knowledgeability, and understanding further highlights an overall cross-cultural consistency (Fig. 3).

While both dimensions showed positive zero-order associations with ratings of wisdom, the magnitude of association (per isomorphic pooled-sample model) was more pronounced for Reflective Orientation, $r = 0.47$, 95% CI [0.46–0.49], compared to Socio-Emotional Awareness, $r = 0.23$, 95% CI [0.22–0.25], $r$(difference) = 0.24, $t = 19.7$, $p < 0.001$. Analogous tests showed a larger divergence in dimensional associations with ratings of targets' knowledgeability, $r$(Reflective Orientation) = 0.50, 95% CI [0.49–0.52] vs. $r$(Socio-Emotional Awareness) = 0.21, 95% CI [0.19–0.23], $r$(difference) = 0.29, $t = 34.6$, $p < 0.001$, and a smaller divergence for ratings of targets' understanding, $r$(Reflective Orientation) = 0.43, 95% CI [0.41–0.44] vs. $r$(Socio-Emotional Awareness) = 0.33, 95% CI [0.31–0.35], $r$(difference) = 0.10, $t = 8.06$, $p < 0.001$.

To examine unique associations of the two dimensions with the attribution of wisdom, we extended the model to a multilevel SEM with

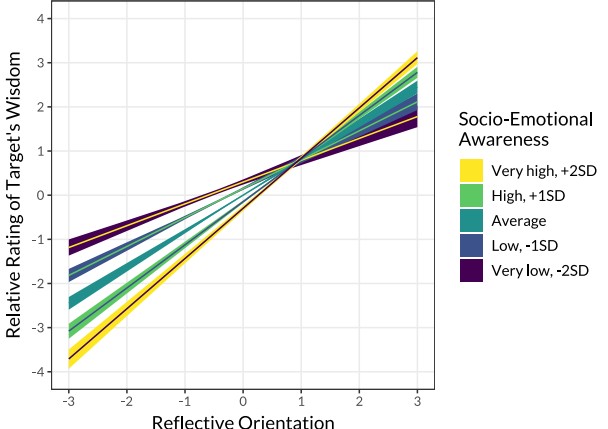

**Fig. 4 | Dimensions of wisdom perception interact in their association with wisdom ratings.** Estimates reflect within-person scores implied by a two-level structural equation model in which two latent predictors, Reflective Orientation and Socio-Emotional Awareness, were allowed to interact in its effect on ratings of targets' wisdom (dependent variable). The bands represent a 95% confidence interval.

a Bayesian estimation where the two latent variables were set to predict wisdom ratings (see Fig. S3). The results showed that Reflective Orientation replicated the positive association with wisdom rating, $b = 0.51$, 95% CI [0.48–0.53], $p < 0.001$, Bayesian credible interval (i.e., a range within which population values fall into with 95% probability) [0.48–0.53], $p < 0.001$, whereas Socio-Emotional Awareness revealed a negative effect, $b = -0.15$ [−0.18 to −0.12], $p < 0.001$. The interaction term, $b = 0.09$, [0.08–0.10], $p < 0.001$, showed that Socio-Emotional Awareness was associated with lower wisdom at the mid- and low levels of Reflective Orientation—lower ratings of wisdom corresponded to lower Reflective Orientation and higher Socio-Emotional Awareness. Conversely, participants gave the highest ratings of wisdom only to the targets they perceived as higher on both latent dimensions (see Fig. 4).

After holding Reflective Orientation constant, we observed that a person was attributed greater wisdom when they appeared less socio-emotionally aware. This pattern was particularly pronounced on the low end of Reflective Orientation. For instance, the scientist and the teacher were perceived as similarly reflective, however the scientist was rated wiser despite the fact that the teacher appeared more socio-emotionally aware. On the other hand, at the higher levels of Reflective Orientation,

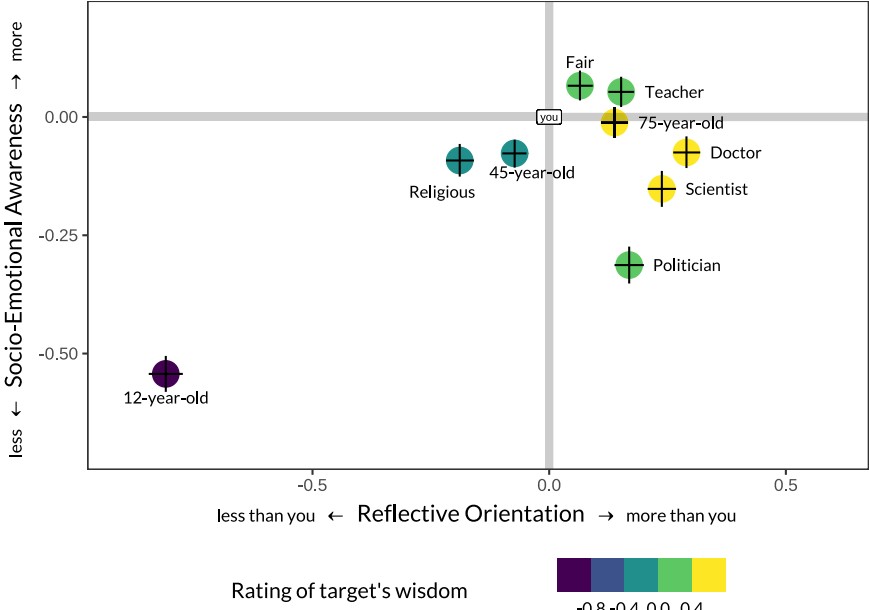

**Fig. 5 | Estimated mean scores of the two wisdom perception dimensions for ten targets.** The dots' position and color represent unstandardized regression coefficients from a two-level pooled-sample structural equation model, CFI = 0.916; RMSEA = 0.027; SRMR$_{within}$ = 0.031; SRMR$_{between}$ = 0.063. Targets were regressed on the two latent dimensions and explicit ratings of wisdom. Values of 'you' were used as a reference category in regression analyses and were therefore set to zero. Thus, all other scores represent the distances from 'you.' The bars represent 95% confidence intervals of estimated parameters for each axis.

Socio-Emotional Awareness was associated with slightly higher wisdom ratings. We will return to this observation in the discussion.

Results were similar for knowledgeability (see Supplementary Note B, Tables S23 and S25). Notably, the role of socio-emotional awareness was more salient for the attribution of understanding—a person had to be high on both dimensions to be considered above the scale midpoint on understanding (see SI Fig. S5). See supplementary results for robustness checks across subsets of targets.

Prior research suggested substantial cross-cultural differences in the attribution of wisdom[4,11]. Our results, in contrast, demonstrate that the associations between the two dimensions and wisdom were surprisingly stable across cultural regions (see SI Tables S24, S26, and Fig. S4). The effect of Reflective Orientation was significant and positive in all eight regions for ratings of wisdom, knowledgeability, and understanding (except for the non-significant effect on understanding in China). The effects of Socio-Emotional Awareness on wisdom ratings were non-significant in North America, South America, and Morocco, but it was significant and negative in the other five regions.

Parallel analyses with raw items (instead of factors) revealed that all 16 items were positively correlated with wisdom ratings across the 6 regions, with only minor exceptions (see Table S28). Further regression analyses with all items as simultaneous predictors of wisdom ratings revealed that in each region the largest significant effects came from 'thinking logically, thinking in many ways', 'applying experiences,' and 'control of emotions'—all part of the Reflective Orientation dimension (see Table S29 in SI).

When examining attribution of knowledgeability, the effects of Socio-Emotional Awareness were negative and significant in all regions but Morocco. Finally, Socio-Emotional Awareness had negative effects on attribution of understanding in India and South Africa, but positive in all the other regions except South America and Morocco where it was non-significant.

**Perception of wisdom in others and the self**

Finally, we compared targets on the latent dimensions of wisdom perception and the explicit ratings of their wisdom (see Fig. 5). As expected, the 12-year-old received the lowest scores on each dimension in each cultural region compared to the other targets. Overall, the doctor and the scientist were the highest on Reflective Orientation, whereas the fair person and the teacher appeared at the top of Socio-Emotional Awareness. An equally high position of the 75-year-old on both dimensions emphasizes the distinctiveness of these dimensions as compared to the ones typically described in the social judgment literature. Consistent with prior research, targets' positions on Reflective Orientation were stable across cultural regions, with average intercorrelation $r = 0.97$, SD = 0.28, whereas targets' positions on Socio-Emotional Awareness were substantially more variable, average $r = 0.81$, SD = 0.28; the mean difference between the targets' positions on the two dimensions across cultural regions, $r$(difference) = 0.16, $d = 0.33$ (see Supplementary Note C for details). Notably, targets with higher social status were rated consistently higher on Reflective Orientation, but inconsistently on Socio-Emotional Awareness. This conceptually replicated the findings from group-based social judgment research, where the perception of agency (a construct similar to reflective orientation) was attributed to hierarchical positions such as social standing, while communion (somewhat similar to Socio-Emotional Awareness) was not[39]. This observation also dovetails with the classic finding in sociology concerning the cross-cultural stability in the perception of competence-based positions of individuals (i.e., occupational prestige)[40].

Researchers from each cultural site picked the gender of the targets deemed culturally appropriate. Therefore, we controlled for the target's gender when examining differences in ratings between targets. Furthermore, we tested how targets' gender is associated with wisdom perception. The results showed that female targets were rated lower than male targets on Reflective Orientation ($b = -0.02$, $p = 0.004$, see Table S30), albeit comparable to male targets on Socio-Emotional Awareness ($b = 0.01$, $p = 0.156$). This result reminds one of the prior research on social judgment and gender stereotypes[41], expanding it beyond the WEIRD samples used in most prior scholarship. Additionally, older and more educated participants showed a weaker method effect, assigning ratings of Reflection Orientation and Socio-Emotional Awareness regardless of the fact whether the target was presented as a single reference or among many comparison characters.

Turning to self-views, participants rated themselves as less reflective compared to six targets, $4.8 < ts \leq 20.5$, but more reflective compared to the religious person, $t = 12.9$, and two non-exemplars of wisdom, the 12-year-old: $t = 51.8$, and the 45-year-old, $t = 5.7$, all $ps < 0.001$, $df = 22,570$. Conversely, participants rated themselves as more socio-emotionally aware than six targets, $4.3 \leq ts \leq 31.0$, $ps < 0.001$, with two exceptions: both the fair person and the teacher were rated as more socio-emotionally aware than the self, $t = 3.3$, 4.1, $ps < 0.001$, whereas the 75-year-old person did not show statistically significant difference from self, $t = 0.7$, $p = 0.461$. Self-ratings were consistent across cultures. At the extreme, participants from Morocco considered themselves to be on the topmost of Socio-Emotional Awareness ($t \geq 4.2$, $p < 0.001$). Parallel analyses with explicit ratings of understanding—a construct invoking socio-emotional abilities—compared to knowledgeability and wisdom yield similar results: Participants in all regions but South Africa (where the difference was not statistically significant, $ps = 0.152$, 0.346) rated their own understanding higher than their knowledgeability, $2.5 \leq ts \leq 13.2$, $p < 0.013$, and wisdom, $4.1 \leq ts \leq 13.7$, $ps < 0.001$ (Fig. S9 in the SI).

## Discussion

In the context of challenging life decisions under uncertainty, people perceived the wisdom of others and themselves along two latent dimensions of mind perception, which concerned how reflective and socio-emotionally aware they perceive the target of judgment to be. These two dimensions were invariant across eight cultural regions representing thirteen languages, thereby extending prior research on mind perception and social judgment beyond the Global North[24]. Our research also squarely focuses on characteristics people attribute to wise decision-making under uncertainty[21,42], in contrast to past research on social judgment about groups or about general mental states. Overall, our results suggest that the structure of the two latent dimensions of wisdom perception is stable across very different cultures, although more work is needed in other parts of the world to comprehensively test whether these two dimensions reflect psychological universals[43].

Three further observations are noteworthy. First, after holding the Reflective Orientation constant, Socio-Emotional Awareness showed a negative association with wisdom ratings. Thus, among equally (less) reflective individuals, targets that were perceived as more caring were viewed as less wise. To elaborate, consider an example of evaluating people who give indiscriminately or people who are mindlessly driven by emotions. These individuals might be admired and revered in some instances, but unlikely to be perceived as wise. Reflective Orientation thus seems to be a necessary condition for obtaining higher wisdom, whereas Socio-Emotional Awareness positively contributes to wisdom only when the first condition is satisfied.

Second, while the cross-cultural agreement about the targets' positions with respect to Reflective Orientation was high[40], we found notable cultural variation in their positions with respect to Socio-Emotional Awareness. Several conjectures may post-hoc explain this observation. One interpretation could involve the grounding of social and emotional acts in local norms, which are more subject to culturally-mediated scripts[44] compared to a more generally applied logic or self-control, at least in the societies examined in the present study. For instance, the attribution of 'care for others' feelings,' one of our 19 wisdom-related characteristics, to doctors might vary more across cultures than the attribution of 'logical thinking.' Another interpretation is that Reflective Orientation may be considered the primary element of wisdom perception across cultures, whereas Socio-Emotional Awareness comes in as a secondary, contextually and culturally dependent element. This conjecture aligns with cultural narratives that often depict wise individuals, such as hermits or philosophers, who, despite their social detachment, are revered for their profound insights into virtuous living. Therefore, while Socio-

Emotional Awareness is an integral aspect of wisdom, its attribution to specific targets, such as professionals or leaders, exhibits greater variability across cultures. This is likely due to its encompassing of a broader range of characteristics, which include both solitary introspection and socially engaged behaviors, leading to diverse cultural interpretations in the attribution of these traits in the context of wisdom.

Third, the two latent dimensions appeared to be differentially susceptible to self-enhancement bias[45]: in most societies, people considered themselves superior to exemplars on socio-emotional competencies while inferior on reflective competencies. The latter observation expands on prior research on personality (i.e., greater self-enhancement of agreeableness versus conscientiousness[46]) and the role of cultural factors such as religiosity for self-enhancement on warmth rather than competence[47]. Contrary to the existing evidence[48], this self-enhancement tendency was present in East Asia as well as in other parts of the world.

In light of previous findings that self-assessments tend to be less accurate when evaluating desirable and behavioral characteristics[49], and that people self-enhance on subjectively-defined traits[50], our results suggest two potential explanations: first, people might value socio-emotional awareness more than Reflective Orientation, leading to greater self-enhancement in this dimension; second, Socio-Emotional Awareness might have a more subjective nature, while Reflective Orientation might point to more directly observed characteristics and 'objective' merits in others. Investigating these possibilities will allow us to refine our understanding of wisdom perception and how individuals may be biased in their assessments. Future research may also explore whether the two dimensions guiding the perception of wisdom also extend to the perception of moral exemplars; if so, it would suggest that moral perception is not a separate domain[51–53], and that evaluation of Reflective Orientation is central for folk theories of morality[54].

Finally, our results might explain why philosophers have long debated whether there are two kinds of wisdom—practical and theoretical wisdom—or whether these two forms of wisdom are in some way unified[55]. The two types of wisdom examined by philosophers may be rooted in the two dimensions of wisdom perception such that theoretical wisdom (or sophia) is more influenced by the perception of characteristics aligned with Reflective Orientation, whereas practical wisdom (or phronesis) is more influenced by the perception of socio-emotional characteristics. Alternatively, Reflective Orientation may be informing both practical and theoretical wisdom, whereas Socio-Emotional Awareness chiefly contributes to the practical wisdom; a fruitful avenue for future research.

Several caveats are in order before concluding. Our study, following the methodologies of prior mind perception[15,28] and social judgment research[24], used a relatively small sample of targets. This decision was rooted in the practical concern that introducing a larger set of targets would have necessitated an unwieldy number of pairwise comparisons. The targets were selected to represent a broad spectrum of common wisdom exemplars[11], ensuring relevance to the study's focus. For the same reason, we allowed for some overlap in target characteristics. For instance, 'you' and a 45-year-old person could also be a teacher. This approach was deliberately chosen to mirror the complexities of real-life social evaluations, where individuals often form judgments based on limited information, and the attributes being compared are seldom mutually exclusive. By adding some context about each target, as well as providing sentence-long descriptions of psychological characteristics, we further expanded our methodology beyond the typical settings in existing research on social and mind perception, where subjects are often described abstractly and limited to one or two words. Our strategy aimed to enhance the ecological validity of our study by introducing relatively more nuanced and context-rich scenarios.

However, this methodological choice also introduces potential biases. The limited number of wisdom exemplars and their overlapping characteristics could have inflated the interdependence of the latent dimensions identified in our analysis. To address this concern, we conducted supplementary analyses excluding age-related targets and self-view ratings. These analyses, along with assessments using random subsets of targets, yielded latent dimensions and degrees of cultural universality similar to our main results, lending robustness to our findings.

Future research in this domain may benefit from expanding the pool of mutually exclusive targets, to reduce the interdependence concern. This expansion, however, must be balanced against the practical challenges inherent in conducting extensive surveys, particularly in societies unaccustomed to such research methods. This balance is critical for ensuring both the feasibility and the comprehensiveness of future studies.

Furthermore, our analyses focused on the most common characteristics used to describe wise persons[21,34,42]. It is theoretically possible that inclusion of more specific behaviors (e.g., praying) or more general psychological attributes (e.g., seeing, feeling, and thinking) would result in further dimensions of wisdom perception. Therefore, it appears prudent to conclude that there are at least two dimensions, which are likely to describe wise persons well.

Moreover, because we relied on convenience sampling, participants were not fully representative of the populations in their respective cultural regions—a common issue for much research in psychology[56]. In addition, the cultures we sampled in our work (e.g., Ecuador and Peru) might not fully represent the broad cultural regions we refer to when we describe our results (e.g., South America). On the other hand, the stability of results across different languages and cultures suggests that the two dimensions of wisdom perception would appear in the broader populations as well. Finally, we aggregated populations that might be heterogeneous (e.g., possible differences between the three South African samples, as well as between Japan and South Korea). We also compared groups that were as different as college students and people from minority populations. We thus cannot entirely exclude the possibility that our aggregative strategy might have obscured some important variation. Further research should examine this question. Whereas the standardized format of our instrument to capture latent dimensions of wisdom perception allowed us to compare wisdom perception systematically across many societies, such questionnaire format may have fostered cross-cultural consistency in participants' reports[57]. Future research might explore whether employing natural-language processing methods to analyze open-ended narratives about wise individuals yields similar cross-cultural consistencies.

Moving beyond delineating wisdom perception dimensions, the next vital step is to assess whether the dimensions we have identified hold the potential for differentiating key competencies in managing life's challenges. A particularly intriguing question for future research is whether people are more likely to trust individuals demonstrating unique features of wisdom in different contexts. For instance, are people more likely to trust individuals they perceive as high in Reflective Orientation in the context of complex problem-solving scenarios while trusting individuals perceived as high in Socio-Emotional Awareness in the context of interpersonal dynamics, such as offering solace in times of sorrow? Investigating these relationships will deepen our understanding of wisdom's nuances and its diverse interpretations across cultures.

## Methods

### Ethics and inclusion

Researchers local to each site provided guidance on the development of materials, data collection, and manuscript preparation stages. Researchers local to each site further confirmed the relevance of materials for their local culture. The collaborator roles were agreed upon at the beginning of the project. All collaborators were included as coauthors. The study was approved by the Ethical Board in each site: by the University of Waterloo Research Ethics Committee in Canada, Ethics Committee of Medical School of Xiamen University in China, Universidad San Francisco de Quito Ethics Committee in Ecuador, IIT Ethics committee in India, Kyoto University Ethics Committee in Japan, Université Internationale de Rabat Ethics Committee in Morocco, Pontificia Universidad Catolica del Peru Ethics Committee in Peru, Ethics Committee of Comenius University in Slovakia, University of Johannesburg Ethics Committee in South Africa, Korea University Ethics Committee in South Korea, The Office of the Human Research Protection Program at the University of California in Los Angeles in USA. Only participants who had signed the informed consent form prior to the study were involved in the research.

### Data

Data was collected between 2019 and 2021 from convenience samples across 16 sites in 11 countries via the Qualtrics online platform or via paper-and-pencil (in Slovakia and Morocco; see Table 2). Samples from Canada, Ecuador, Peru, and the US consisted of university students, the other samples came from a broader population. Japan and two Indian samples used the shortened version of our questionnaire (limited to five targets). Based on the GPower calculation for 80% power

**Table 2 | Sample characteristics and correlations between reflective orientation and socio-emotional awareness**

| | r | CI 95% | Age mean (SD) | % Female | N | Languages |
|---|---|---|---|---|---|---|
| *Asia* | | | | | | |
| India | 0.88 | [0.84 –0.92] | 30.9 (10.9) | 50 | 374 | Hindi, Tamil, and Meitei |
| China | 0.84 | [0.79– 0.88] | 22.6 (5.9) | 71 | 225 | Mandarin |
| Korea and Japan | 0.75 | [0.69– 0.81] | 42.3 (0.5) | 50 | 308 | Korean and Japanese |
| *Africa* | | | | | | |
| South Africa | 0.83 | [0.78– 0.87] | 34.5 (11.9) | 64 | 524 | Afrikaans, Sepedi, and Zulu |
| Morocco | 0.33 | [0.24– 0.43] | 34.4 (14.2) | 47 | 181 | Arabic |
| *Europe* | | | | | | |
| Slovakia | 0.77 | [0.73– 0.81] | 30.1 (13.2) | 24 | 246 | Slovakian |
| *Americas* | | | | | | |
| North America (Canada and US) | 0.58 | [0.52– 0.63] | 26.7 (10.5) | 64 | 500 | English |
| South America (Ecuador and Peru) | 0.41 | [0.32– 0.49] | 22.3 (4.8) | 59 | 349 | Spanish |
| *Pooled sample* | 0.69 | [0.66– 0.71] | 30.6 (12.7) | 55 | 2707 | |

*Note: r = correlation coefficients, estimated at the within level by a partial metric invariance multigroup multilevel confirmatory factor analysis model (see caption to Fig. 2 for details).*

and small effect size ($r = 0.21$), we required a sample of at least 173 participants per site. Notably, our study involved samples from indigenous and rural groups from several societies (e.g., Meitei people in India), for which it was not feasible to obtain larger samples. We aimed for at least 100 from indigenous and minority groups and at least 180 participants from larger populations. The collected total sample consisted of 2650 participants.

## Sample composition

For pragmatic reasons, we relied on convenience samples. Consequently, sample characteristics varied across sites, allowing us to test our measurement model of wisdom perception across regions differing widely in age, socio-economic status, and religiosity. As reported in Table 2, the average age was around early 20s for samples in North and South America and China, around 30 in Africa, Slovakia, and India, and 42 in Korea and Japan. The average age in a pooled sample was 30. In India, Korea, Japan, and Morocco, half of the participants were female, whereas in the Americas and South Africa, the share of female participants was somewhat higher than 50%. In China, most participants were female (71%), whereas in Slovakia they were a minority (24%).

Samples also varied in socio-economic status and religiosity. Some participants were undergraduate students; therefore, level of education was measured for the participant's parents. Parental education was lower in India, Korea & Japan, Slovakia, where most of participants (>50%) had parents with education below college. Parents of the South American participants were the most educated (60% had higher education), whereas North America, South Africa, and China were in-between (Table S4 in SI).

The religiosity of participants varied widely, too (Table S4). Participants reported the importance of religion in their lives on a 5-option scale from 'Not at all' to 'Very important.' The proportion of participants reporting that religion is 'Very' or 'Quite' important in their lives varied from 80% in Morocco, to 77% in South Africa, to 42% in India, to 40% in Slovakia, to 32% in South America and 22% in North America. For pragmatic reasons, we did not measure religiosity in China.

## Sample size considerations

For technical reasons, we failed to pre-register the methods prior to data collection, albeit approving the method internally by the Geography of Philosophy consortium (see unedited copy on OSF). Following prior work on mind perception[15], initially, we planned to use multidimensional scaling (MDS) and had pragmatic concerns for samples in harder-to-get populations. Thus, we estimated a minimal sample size per group (targeted at 180 with a minimum of 100 for smaller populations). However, in the end, we decided to employ a more advanced technique involving multilevel structural equation modeling. This analytical procedure is conceptually similar to MDS. However, it has critical advantages concerning control for several sources of potential biases (e.g., nested structure of the data, ability to simultaneously estimate latent variables and their direct impact on the dependent variable, and ability to estimate measurement error).

Notably, this method called for larger samples. Therefore, we merged smaller samples into eight broad cultural regions based on broadly applied classifications of cultural similarity in values, practices, and relational and self-concepts[32,33,56]. First, we merged samples taken from the same countries (e.g., three linguistic samples in South Africa were treated as one). Second, we followed a widely consensual classification of countries to merge American and Canadian samples into the North American group and Ecuador and Peru into the South American group. Here, we followed prior insights on cultural values[32] and relational and self-views (ref. 33. for a review). We further combined South Korean and Japanese samples, because the two countries are the wealthiest in the East Asian region, with common features of

economic and political systems, as well as some cultural features[32,33]. Moreover, Japanese participants completed only a subset of targets, and the sample on its own was severely underpowered for the multilevel SEM models. We treated the Chinese sample as distinct due to the special position it takes in the region and its distinct socio-economic system. We treated samples from Morocco and Slovakia as sole representatives of their cultural regions (North Africa and Europe) and did not merge them; these samples also varied in modality (paper-and-pencil versus online in other sites).

## Procedure

After providing the informed consent form, participants compared pairs of human targets in regard to their likeliness to employ each of the 19 ways to deal with a difficult life situation (see Fig. 1). First, participants were presented with one of the pairs of targets. Each target had a culturally specific name and a short description that contextualized their exemplary qualities. For instance, instead of simply stating 'teacher,' participants read 'Dr. Kim is a schoolteacher who educates 12-year-olds about local history and literature.' Similarly, instead of 'scientist', we provided a concrete description 'Dr. Kim is a scientist who gathers information about plants, animals, and people to make sense of the world' (see details in Supplementary Methods in SI).

To reduce study fatigue, participants were randomly assigned to only one reference target from the list of ten (between-subject element), to which they compared all other targets (within-subject element; presented in a pseudo-random order). Thus, participants saw individual pairs constructed between that reference target and each of the nine remaining targets. Second, participants responded to a key question: 'How likely is it that [reference target] will do the following things compared to [comparison target]?' when they 'are trying to make a difficult choice that there is no clear right or wrong answer to.' The choice of pairwise comparisons was meant to control for the individual general response preferences prevalent in surveys with rating scale questions[58]. Moreover, survey response style differs across cultures and, therefore, can bias the results of the cross-cultural comparisons (e.g.,[59]). Comparison criteria consisted of 19 characteristics, such as 'think about the issue in many different ways' and 'have good control of emotions' (see exact wordings in SI Table S3). Participants compared targets using a five-point scale from 'Much less likely' to 'Much more likely' with a middle option 'Equally likely.' Subsequently, participants provided ratings of each target's wisdom, knowledge, and understanding (in a randomized order) using a five-point scale from 'Not [wise] at all' to 'Extremely [wise].' We also collected basic demographic information such as age, gender, and education of participants.

The initial version of the instrument was developed in English by an international team of researchers representing different cultures in the sample. Translating philosophical terms is difficult due to a range of epistemological traditions across cultures[5]. A notable issue in this study was the inherent ambiguities in the English terminology, exemplified by the word 'understanding,' which can signify either comprehension of information or empathy towards others' feelings. Such ambiguities might not have direct equivalents in the target languages or when present, could function differently, as seen in the nuanced usage of 理解 in Chinese. To address these complexities, our approach was twofold. Firstly, we enlisted an extensive team of experts in cross-cultural research, linguistics, and anthropology. These specialists played a crucial role in the translation, adaptation, and validation of all materials. Secondly, in-depth discussions were conducted focusing on key terms like wisdom, knowledge, and understanding, ensuring cultural appropriateness and semantic accuracy. This process involved classical back-translation techniques and consensus-building among experts to finalize the terms used across different cultural sites. Additionally, research teams at each site were encouraged to report any challenges encountered during the translation of philosophical

terms, facilitating team deliberations to resolve ambiguities and align interpretations in the target language.

**Materials.** A total of ten targets were included in the study: self, scientist, doctor, teacher, fair person, politician, religious person, 75-, 45-, and 12-year-old (see wordings in Supplementary Methods). The selection of characters was performed by an interdisciplinary group of experts and followed three criteria:

(1) The target is an exemplar of wisdom (as evidenced in the literature; see[11]), with two control targets—a 12-year-old person who commonly does not possess much life experience and a 45-year-old person as a representative of an average adult in many societies.

(2) We expected each selected target to be understandable and common in each of the sampled societies.

(3) The final list of targets would be reasonably small to enable pairwise comparisons without fatiguing the participants; this is because each new target would involve 19 extra comparisons, and with the nine comparison targets, it counted up to 171 comparisons for each participant.

We generated 19 characteristics following the core items from the previously established common wisdom model[21] (also see[22]), and similar frameworks featuring additional characteristics such as emotion regulation[42]. These items included meta-cognitive characteristics, as well as prosocial features such as cooperation. Further, we included two items referring to attention to one's and others' bodily expressions, based on the idea that wisdom is associated with mindfulness[42,60] and that attention and bodily awareness are central elements of mindfulness[61]. This way, we sought to accommodate evidence from prior cross-cultural scholarship suggesting that in non-WEIRD countries, wisdom may be aligned with social, spiritual (or nature-related), and visceral experiences[4]. Together, these characteristics corresponded to the dimensions of mind, heart, and body discussed in prior mind perception research[16,28]. To increase variance in the data, we also added one (reverse-coded) item concerning the lack of humility—showing pride in oneself, as well as one evolutionary adaptive, but arguably morally ambiguous feature concerning in-group favoritism. Each characteristic described a behavior, a mental action, or a focus of attention and did not explicitly refer to wisdom.

Participants compared targets along these 19 characteristics using a five-category scale from '[the reference target] is much less likely than [the comparison target]' to '[the reference target] is much more likely than [the comparison target].' The five-category scale allowed for

the neutral option where the two characters were equally likely to perform a given action.

**Data transformations.** Missing values were treated with the full information maximum likelihood method, which makes use of all the available information when estimating a model. Since the measurement instrument could not include comparisons of targets to themselves, we completed the data by filling it in with response category 3 ('equally likely'). For the within-individual level analysis, answers to the questions about how wise, knowledgeable, and understanding each target were transformed into differences between each pair of targets involved in the comparisons. For instance, if the reference target for the respondent was a teacher, participants were supposed to compare the teacher to every other target using each of the 19 behavioral items. Correspondingly, we computed the difference between the wisdom rating of the teacher and the wisdom of each of the targets given by the respondent (see the structure of the data in Table S5). All the variables were centered around the grand mean for all analyses. For the analysis of the pooled sample and region-specific models, we standardized all the variables (subtracted mean and divided by their standard deviation).

**Analytical approach.** The analytical strategy followed multiple steps within the multilevel analysis framework (see overall structure in Fig. 1). Our multilevel model included two levels, one of which represented within-individual structure and the other reflected between-individual differences (Fig. 6). The design of the instrument implied that the within-individual level described the comparisons of a single reference target with many comparison targets, whereas the between-individual level described the comparisons of various reference targets to the individual averages of all the comparison targets. At the within-level, a reference target was constant, because each participant had only one reference target. Thus, only the comparison targets contributed to the within-individual variance. By virtue of the randomization of the reference targets, it was reasonable to expect that structures and associations between variables would be similar across the two levels. We used this two-level structure to fit exploratory and confirmatory factor analyses and then extended it to the multilevel structural equation models to test latent dimensions' associations with explicit attributions of wisdom. The latter controlled for individual differences, including age, gender, education, and religiosity.

Following preliminary analyses, we excluded three items from further consideration: One item ('disengage from the situation and let it unfold as it does') conceptually deviated from the others because it

A. Conceptual representation of the measurement model

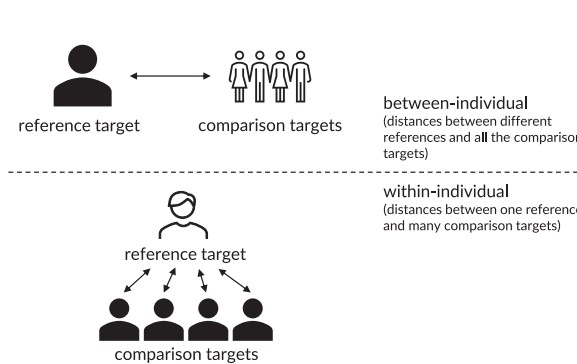

B. Diagram of multilevel factor analysis

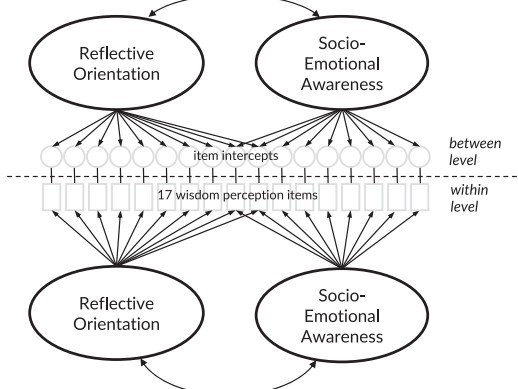

**Fig. 6 | Conceptual and statistical representation of the multilevel measurement model of wisdom perception. A** At the within-individual level, different comparison targets are related to a reference target; at the between-individual level a reference target is related to an aggregate of all comparison targets. **B** The items

are wisdom-related characteristics; at the between-individual level, they are represented by their intercepts. Factor indicators at both levels have residuals that are not shown. The factor loadings were constrained to be equal across the levels (i.e., isomorphic).

uniquely implied inaction, while the task given to the participant was to determine if a target 'will do' something. It also showed negligible associations with either factor (loadings < 0.2, see Supplementary Methods, Tables S6 and S7). Another item ('show pride in themselves') was the only reverse-coded characteristic (opposite of humility), and thus inconsistent with others. Finally, an item 'notice if their body tenses up or relaxes when thinking about different options' was associated with each factor but did not contribute to the content coverage. Key analyses in the main body of the manuscript yield largely identical results when performing analyses on all 19 items. Thus, we focus on the restricted set to ensure clarity and avoid possible bias due to a single reverse-coded item or items that stand out from the others.

To test measurement invariance, we expanded the two-level confirmatory factor model to the multiple-group-multilevel factor model[62]. Here, we fitted the original multilevel model simultaneously in several groups. This approach allows for checking whether the parameters of multilevel models are similar across subpopulations. Due to the structure of our data, which is based on pairwise comparisons, within-level intercepts were naturally zero. It means that we were able to test only for configural and metric invariance. The configural model did not constrain factor loadings (except for those used for model identification). Since we opted for an isomorphic model, the factor loadings were constrained to equality across levels within each group separately, that is, loadings were similar within groups in all models, including configural. Metric invariance model constrained factor loadings to equality across groups. Combined with isomorphism, it resulted in a model that constrained the factor loadings both across levels and between groups. A small difference in the fit between the configural and the metric invariance models was considered evidence of the invariance per Chen's criteria[63].

Most models were fitted using a maximum likelihood robust estimator, and the models with the interaction terms between the latent variables used Bayesian estimation. We labeled models that did not account for group differences as 'pooled sample' models. We took into account the fact that participants came from different cultural regions and, therefore, were drawn from several rather than a single population by correcting standard errors and fit statistics in the pooled sample models for clusterization[64]. All the statistical tests reported are two-sided. All the models were run in Mplus 8.8 software[65]. Summaries and extra analyses were run within R 4.1 environment[66] and made use of over 20 packages listed in the provided script.

### Reporting summary

Further information on research design is available in the Nature Portfolio Reporting Summary linked to this article.

## Data availability

The full data are available at an OSF directory https://doi.org/10.17605/OSF.IO/M4DXV.

## Code availability

The R and Mplus codes to reproduce all the data analyses, figures, and tables are available at an OSF directory https://doi.org/10.17605/OSF.IO/M4DXV.

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

## Acknowledgements

The authors are grateful to Rebecca Neel for her comments on the earlier version of this paper. H. Clark Barrett, Badr Guennoun, Edouard Machery, Veli Mitova, Josien Reijer, Pedro P. Romero, and Stephen Stich were supported by the John Templeton Foundation, grant 60813; Igor Grossmann was supported by Social Sciences and Humanities Research Council of Canada Insight Grant 435-2014-0685, and John Templeton Foundation grant 62260.

## Author contributions

Conceptualization: H. Clark Barrett, Wesley Buckwalter, Edouard Machery, Martin Kanovský, and Igor Grossmann. Data curation: Maksim Rudnev, Daniel A. Wilkenfeld, Kelli Barr, Emanuele Fabiano, Ari D. Fodeman, Júlia Halamová, Joshua Homan, Martin Kanovský, Minha Lee, Xiaofei Liu, Ljiljana Pantovic, Josien Reijer, Pedro P. Romero, and Lixia Yi. Formal analysis: Maksim Rudnev, Ari D. Fodeman, and Igor Grossmann. Funding acquisition: H. Clark Barrett, Edouard Machery, Stephen Stich, and Igor Grossmann. Investigation: Kelli Barr, Abdellatif Bencherifa, Rockwell F. Clancy, Yasuo Deguchi, Emanuele Fabiano, Badr Guennoun, Júlia Halamová, Takaaki Hashimoto, Joshua Homan, Kaori Karasawa, Hackjin Kim, Jordan Kiper, Minha Lee, Xiaofei Liu, Pablo Quintanilla, Josien Reijer, Pedro P. Romero, Salma Tber, and Igor Grossmann. Methodology: Maksim Rudnev, Edouard Machery, Ari D. Fodeman, Martin Kanovský, and Igor Grossmann. Project administration: Edouard

Machery, Stephen Stich, Daniel A. Wilkenfeld, Kelli Barr, Abdellatif Bencherifa, Rockwell F. Clancy, Yasuo Deguchi, Emanuele Fabiano, Badr Guennoun, Júlia Halamová, Joshua Homan, Martin Kanovský, Kaori Karasawa, Hackjin Kim, Jordan Kiper, Minha Lee, Xiaofei Liu, Veli Mitova, Rukmini Bhaya Nair, Ljiljana Pantovic, Pablo Quintanilla, Josien Reijer, Pedro P. Romero, Purnima Singh, Salma Tber, and Igor Grossmann. Resources: Wesley Buckwalter, Daniel A. Wilkenfeld, Kelli Barr, Abdellatif Bencherifa, Veli Mitova, Rukmini Bhaya Nair, Pedro P. Romero, Purnima Singh, and Igor Grossmann. Software: Maksim Rudnev, Daniel A. Wilkenfeld, Pedro P. Romero, and Lixia Yi. Supervision: Edouard Machery, Abdellatif Bencherifa, Badr Guennoun, Martin Kanovský, Hackjin Kim, Veli Mitova, Rukmini Bhaya Nair, Pablo Quintanilla, Purnima Singh, and Igor Grossmann. Validation: Maksim Rudnev and Igor Grossmann. Visualization: Maksim Rudnev and Igor Grossmann. Writing - original draft: Maksim Rudnev and Igor Grossmann. Writing - review & editing: Maksim Rudnev, Wesley Buckwalter, Edouard Machery, Stephen Stich, Daniel A. Wilkenfeld, Abdellatif Bencherifa, Damien L. Crone, Badr Guennoun, Júlia Halamová, Joshua Homan, Martin Kanovský, Veli Mitova, Ljiljana Pantovic, Brian Porter, Josien Reijer, Pedro P. Romero, Lixia Yi, and Igor Grossmann.

## Competing interests

The authors declare no competing interests.
