## [Peer Review File · Nature Communications]

Reviewers' Comments:

Reviewer #1:

Remarks to the Author:

This paper explores whether there are any latent dimensions underlying the social perception of wisdom that are common across cultures. An exploration of samples from 11 countries revealed that two dimensions emerged in each of the cultural regions: reflective orientation and socioemotional awareness. Curiously, the two dimensions had an interactive effect such that a reflective orientation contributed to wisdom judgements, whereas socioemotional awareness only positively contributed to wisdom perceptions among those who were high in reflective orientation.

I think this paper makes an important contribution to the growing literature on wisdom. While people seem to have an intuitive understanding of what is wise, it is unclear how similar their understandings are across cultures. The researchers used a sophisticated technique of comparing a series of ten individuals on 19 separate characteristics regarding wise decisions. While there is much literature that would have predicted cultural variability in what is considered wise, the present research found evidence for striking similarities in the basis of wisdom across cultures, although the cultures did vary importantly on how central the socio-emotional dimension was to wisdom judgments. That the research was conducted across such diverse regions of the globe makes these findings all the more impressive.

The authors combined samples that were very different, such as college students and people from indigenous populations, and samples from separate countries, such as Korea and Japan. This combining was necessary because their individual sample sizes, especially from the indigenous populations, were too small to analyze using the methods here. I think the authors should more explicitly call attention to the limits of combining these different samples. On the one hand the findings looked quite similar across the different world regions, but it's hard to say this with much precision given the combining that was done. I think the authors should highlight that their combining of quite different samples could potentially obscure some important cross-cultural variation.

The fact that the cultures varied more in socioemotional dimension than in the reflective dimension is interesting. Wisdom seems to be more universally about reflection, whereas cultures differ in how much socioemotions are involved. Why do the authors think this may be the case? At present they don't really offer much thought on this. I think they should offer more explanation for this difference between the dimensions.

It is interesting that people rated themselves better than others especially for socio-emotional awareness. The authors did not discuss what seems to me to be the more likely explanation that socio-emotional traits are less objective, and thus easier to view oneself positively. People tend to self-enhance more on subjectively defined traits. See Dunning, Meyerowitz, & Holzberg, 1989 JPSP.

The caveats that the authors listed here are important, and they do limit the generalizability of these findings. I think the authors should emphasize these limitations more in their discussion of their findings earlier in the paper. As it currently reads, the results sound far more conclusive until one gets to the caveats section.

Reviewer #2:

Remarks to the Author:

Dimensions of Social Perception of Wisdom Across Five Continents

General thoughts:

Conceptually, this is a really interesting paper and is very well written. It's particularly noteworthy that despite different epistemological traditions across cultures, people tend to agree on who is wise and which traits constitute wisdom. The authors should be applauded for their largescale cross-cultural approach to tackling the social perception of wisdom.

This being said, I struggle with this paper's unique contribution to the social perception literature. It is somewhat unclear whether the two dimensions of wisdom perception differ meaningfully from dimensions of social perception identified in previous literature, nor whether these dimensions are uniquely capturing wisdom perceptions or some other related construct. Finally, the interaction effect between the two dimensions, while paradoxical and intriguing, merits further explanation and may actually challenge the authors' claim that wisdom perceptions vary along two dimensions.

These issues, along with several miscellaneous points, are outlined in greater detail below.

The Two Dimensions of Wisdom Perception

Core to the authors' claim is that perceptions of a target's wisdom vary along two dimensions: Reflective Orientation and Socio-Emotional Awareness. The authors acknowledge that these two dimensions are similar to previous work that has divided social perception along the dimensions of agency (or competence) vs experience (or warmth). But, looking at the items included in each respective dimension, it is unclear whether these new dimensions are distinct from the dimensions already described in the social perception literature. Reflective Orientation is comprised of control/logic/acting and Socio-Emotional Awareness is comprised of feeling/caring. The labels of the dimensions are of course just that, labels, and I think it's possible that the dimensions of wisdom perception are the same as the dimensions of social perception more broadly, in which case the question arises of whether this is revealing that universal dimensions of person/social perception (Susan Fiske's work) apply for judgments of wisdom, which are social perceptions. If warmth/feeling and competence/thinking are universal dimensions, then the authors might argue why it is novel to reveal them for judgments of wisdom.

The Uniqueness of Wisdom Perception

One way to increase the novelty of these findings could be to show that the two dimensions pertaining to wisdom perception are uniquely associated with perceptions of wisdom, but they don't appear to be. Each dimension was positively correlated with explicit ratings of wisdom, as we would expect them to be, but the dimensions also showed similar correlations with explicit perceptions of knowledge and understanding. So again, are these dimensions uniquely capturing perceptions of wisdom, or some broader construct like intelligence or decision-making ability? If the authors are arguing this is special to wisdom, I think they need to do more to demonstrate that these dimensions are really capturing wisdom perception. Given the results, it feels like you might be able to replace "wisdom" with "knowledge" or "understanding" throughout the paper, even though each of these has different connotations. The fact that the two dimensions were not uniquely correlated with wisdom judgements could also be further evidence that they reflect the two universal dimensions of social perception?

Puzzling Interaction Effect

The most interesting finding in the paper was the interaction effect between the two dimensions predicting wisdom, and this seems to be where the paper diverges from other social perception research. It's certainly puzzling that at the high end of reflectiveness, wiser people are those who have higher socio-emotional awareness, but that at the low end of reflectiveness, wise people have lower socio-emotional awareness. This leads to paradoxical patterns of results, like scientists being wiser than teachers despite having less socio-emotional awareness.

Given this finding, the two current dimensions seem to not fully capture how people think about wisdom. If someone of equal reflectiveness can have less socio-emotional awareness and yet be wiser, then there must be something else that is driving perceptions of wisdom. To make this more explicit, we can consider the case of mind perception, which is analogized in the manuscript. Mind perception varies along the two dimensions of agency and experience. Each dimension is positively associated with perceived mind: more agency  more mind at every level of experience, and more experience  more mind at every level of agency. But imagine that this research had instead found that having more agency sometimes leads to lower perceptions of mind. This would be a puzzle about perceived mind that demands an explanation.

Miscellaneous Points

I think it would be useful to more explicitly define Reflective Orientation and Socio-Emotional Awareness in the intro, as opposed to just in the methods/results, since so much of the interpretation hinges on how we understand these dimensions. Maybe list a few more items that went into them, or just in simple terms how we might think of them.

Similarly, I think it would be useful to provide a bit more detail in the intro about how people rated targets' wisdom. Were the pairwise comparisons dichotomous (e.g., scientist more logical than teacher) or continuous? Did the prompt ask participants to think about the targets in the context of a specific decision, or a difficult decision in general? These questions were answered later in the paper but were unclear the first time the measure was introduced.

The word "dovetails" is used heavily – it might improve flow to vary the word choice, using terms like converges with/supports other research, etc.

In the first paragraph of the discussion, the phrase "our results suggest that the two latent dimensions of wisdom perception may be a psychological universal" feels like quite a strong claim. I feel like the authors can walk this back a bit and say that the effects were stable across cultures.

Concluding thoughts

In sum, I think the authors tackled an interesting question and I really appreciate the cross-cultural data and the elegant descriptive analyses. My uncertainty about the merits of the manuscript for Nature Comms revolves around the novelty. Just finding two dimensions—if they are the same universal dimensions of person/social perception—may not be sufficiently new especially if ratings of wisdom are the same as other constructs like knowledge. At the same time, there is clearly a wrinkle in wisdom ratings because of that interaction. That interaction might be the key to demonstrating novelty, but it is not sufficiently unpacked or explained in the current version.

Another thing that might help is to think about the behavioral consequence of these dimensions. It seems like one reason that the warmth/competence work got so much traction is because it connected to behavior and other judgments (e.g., warmth is friend/foe). As the wisdom research line moves forward, I think it will be useful to consider not only whether Reflective Orientation and Socio-Emotional Awareness are unique from things like warmth and competence, but also whether each might have unique behavioral judgments. Are people more likely to trust the wisdom of certain dimensions in certain settings?

Reviewer #3:

Remarks to the Author:

This is a very interesting paper that aims to use an innovative methodology to identify a cross-cultural core of conceptions of wisdom. I applaud the authors for the immense effort invested in

this study. At the same time, I have some concerns about several aspects of the methodology. I will try to explain these concerns in the following. As I am not familiar with some of the complex methods used for data analyses, I focus on the substantive meaning of the findings rather than on the technical decisions the authors made in analyzing the data.

First of all, the study involved ten different target characters: oneself, a 12-year-old, a 45-year-old, a 75-year-old, a religious person, a fair person, a teacher, a politician, a scientist, and a doctor. I am not sure I understand the rationale underlying this selection of characters. In the supplementary information, the authors say that “we selected a teacher, a scientist, and a doctor, because they are explicitly mentioned as exemplars of wisdom in prior North American literature (e.g., Weststrate et al., 2016). Finally, we selected a politician and a fair person (who is an activist fighting for human rights) as exemplars of civic leadership—another domain often associated with wisdom (Weststrate et al., 2016).” (page 3). So was the goal to select targets that participants would consider as wise? Then why not just use one target representing the wisest person participants know? In the Weststrate et al. study, some politicians (e.g., Lincoln, Churchill) were nominated as wise, but the target description might just as well make participants think of Donald Trump or their corrupt local mayor. Not every politician is considered as wise, nor is every teacher, doctor, or scientist. Also, participants in Japan and India were only presented with the targets “self,” “religious person,” and the three age groups, that is, with zero wisdom exemplars, so I’m not sure how much the data contributed to the final analyses.

Anyway, each participant was asked to compare one (randomly assigned) target to all of the other targets with respect to how likely they would show a total of 19 behaviors or ways of thinking in a difficult life situation. These characteristics were selected, according to the authors, because they are “associated with wisdom in prior philosophical and psychological scholarship on wisdom” (manuscript, p. 2). I am not sure which philosophical or psychological scholarship suggests that “maximizing the benefit for one’s group, regardless of the cost for others,” “not showing emotions,” or “paying attention to what nature tells you” is highly characteristic of wisdom. Second, it seems that the characteristics are mostly representative of Western conceptions and those non-Western conceptions that have already been investigated. If the concept of wisdom in Japan or South America were very different from those in North America and Western Europe, how would this study detect that, given that neither the targets nor the characteristics would then be representative of wisdom in those parts of the world?

Participants provided their comparisons on five-point scales ranging from “much less likely” to “much more likely” – e.g., they would judge whether their target, the doctor, was less or more likely to think about the issue in many different ways than a 12-year-old. While this method provides an impressive number of data points, I did not really understand what its advantages are in comparison to simply asking participants to rate the different targets on the different characteristics. Which potential biases or distortions are eliminated by using pairwise comparisons instead, especially when the comparisons are between different figures that are all assumed to be (relatively) wise? In other words, why does information on how participants in different cultures describe a doctor in comparison to, say, a religious person tell us more about their conceptions about wisdom than just looking at how they describe a doctor (or perhaps simply how they describe a very wise person)? It would be great if the authors could explain that in the paper.

After completing a rather long list of comparisons, participants were also asked to rate all 19 targets with respect to wisdom, understanding, and knowledgeability (in addition to the translation difficulties with the word “understanding” that the authors mention, it would also be interesting to know how they ensured the comparability of the words for “wisdom” in the different languages).

Analyses were highly complex because of the complex data structure, but they showed that quite universally, the 19 characteristics grouped into two factors. This does not seem particularly surprising given that the authors themselves say that other research has already shown the cross-cultural existence of two cardinal dimensions of social judgment. Looking back at the target

characters listed above, it makes a lot of sense that many of them (scientist, doctor, teacher) would be considered as high in the Reflective Orientation but not necessarily the Socio-Emotional Awareness factor. That might also explain why participants in most regions considered themselves as higher than the targets in Socio-Emotional Awareness but not in Reflective Orientation.

The strongest evidence for the 19 characteristics indeed being representative of wisdom comes from the correlations between participants' wisdom ratings of the various targets and their judgments of the 19 characteristics. If participants rated a target as wiser, they also rated that target as more likely to show the behaviors on the Reflective Orientation factor and, to a lower extent, the Socio-Emotional Awareness factor. Basically, it seems to me that these findings show that participants rated, say, a doctor as wiser than a 45-year-old (see Figure S7) and that they also considered the doctor as higher in at least the Reflective Orientation factor. I'm not sure that the findings provide strong evidence that the characteristics represented by the two factors are the key characteristics that people across the world consider as typical for wisdom. There seems to be some circularity in the fact that the researchers selected the characteristics from existing conceptions of wisdom, then showed that they are indeed correlated (one of them not even strongly) with ratings of wisdom.

I do think there is some highly interesting evidence in this data set, but some of it may be in the differences between regions rather than in the commonalities. For example, I found it very interesting that nature and divinity had higher loadings on the Socio-Emotional Awareness factor in traditional communities. Also, did the wisdom ratings of the different targets vary between regions? For example, did participants everywhere consider the religious person as equally unwise? Did such variations perhaps go along with differences in characteristics ascribed to the targets? In addition, there seem to be considerable differences in sample compositions across regions, especially with respect to education, which should be documented in more detail in the supplemental materials. For example, why were there so many male participants in the Slovak sample? To what extent do differences in participant age, gender, or education account for some of the regional differences? (I noticed that the Slovak sample had the lowest correlation between Socio-Emotional Awareness and wisdom ratings, and Table S23 seems to suggest that this may be a gender effect.) I understand that it is probably difficult to look for differences in this rich trove of data without clear theory-based hypotheses, so there may be other ways to enhance the incremental value of this study for our knowledge about cultural differences in wisdom – right now, I am not quite sure how much we can really learn from it.

Small issues:

Page 3: "Contrary to our expectations, these dimensions were consistent across cultures:" what were the authors' expectations and why?

Page 8: "in real life people typically evaluate others based on limited information about them:" I do not quite see that providing one sentence of description increases ecological validity – in comparison to what?

Reviewer #4:

Remarks to the Author:

Based on the statistical review of the manuscript, it seems that the authors do an excellent job explaining the multi-level analysis results to the domain expert. The explanations of the factor analysis results, particularly, were very clear and precise. It is evident that the authors have a good understanding of this concept. The figures, especially factor loadings (Figure 2), were also found to help clarify the analysis. Furthermore, the code used for the analysis was found to be precise and reproducible.

However, the reviewer recommends that the authors clarify in the text that a Bayesian approach was used. Although it is clear from the code that Bayesian analysis was used, nothing in the text suggests it. It would be beneficial to comment on what prior was used for which parameters and provide an explanation suggesting Bayesian analysis. Overall, the statistical analysis of this manuscript was found to be correct.

Response to Reviewers

Reviewer #1

This paper explores whether there are any latent dimensions underlying the social perception of wisdom that are common across cultures. An exploration of samples from 11 countries revealed that two dimensions emerged in each of the cultural regions: reflective orientation and socioemotional awareness. Curiously, the two dimensions had an interactive effect such that a reflective orientation contributed to wisdom judgements, whereas socioemotional awareness only positively contributed to wisdom perceptions among those who were high in reflective orientation.

I think this paper makes an important contribution to the growing literature on wisdom. While people seem to have an intuitive understanding of what is wise, it is unclear how similar their understandings are across cultures. The researchers used a sophisticated technique of comparing a series of ten individuals on 19 separate characteristics regarding wise decisions. While there is much literature that would have predicted cultural variability in what is considered wise, the present research found evidence for striking similarities in the basis of wisdom across cultures, although the cultures did vary importantly on how central the socio-emotional dimension was to wisdom judgments. That the research was conducted across such diverse regions of the globe makes these findings all the more impressive.

Response: Thank you so much!

The authors combined samples that were very different, such as college students and people from indigenous populations, and samples from separate countries, such as Korea and Japan. This combining was necessary because their individual sample sizes, especially from the indigenous populations, were too small to analyze using the methods here. I think the authors should more explicitly call attention to the limits of combining these different samples. On the one hand the findings looked quite similar across the different world regions, but it's hard to say this with much precision given the combining that was done. I think the authors should highlight that their combining of quite different samples could potentially obscure some important cross-cultural variation.

Response: We fully agree and now acknowledge the limitations related to the combining of the samples and heterogeneity of samples, P.14:

“...we aggregated populations that might be heterogeneous (e.g., possible differences between the three South African samples, as well as between Japan and South Korea). We also compared groups that were as different as college students and people from indigenous populations. We thus cannot entirely exclude the possibility that our aggregative strategy might have obscured some important variation. Further research should examine this question.”

The fact that the cultures varied more in socioemotional dimension than in the reflective dimension is interesting. Wisdom seems to be more universally about reflection, whereas cultures differ in how much socioemotions are involved. Why do the authors think this may be the case? At present they don't really offer much thought on this. I think they should offer more explanation for this difference between the dimensions.

Response: Although these could be a little speculative, we agree that giving some idea of the meaning of the results is important.

We now added our interpretations, page 12:

“The cross-cultural agreement about the targets’ positions with respect to Reflective Orientation was high (cf. Treiman, 1977), we found notable cultural variation in their positions with respect to Socio-Emotional Awareness. Several conjectures may post-hoc explain this observation. One interpretation could involve the grounding of social and emotional acts in local norms, which are more subject to culturally-mediated scripts (Lindquist et al., 2022) compared to a more generally applied logic or self-control, at least in the societies examined in the present study. For instance, the attribution of “care for others’ feelings,” one of our 19 wisdom-related characteristics, to doctors might vary more across cultures than attribution of “logical thinking.” Another interpretation is that Reflective Orientation may be considered the primary element of wisdom perception across cultures, whereas Socio-Emotional Awareness comes in as a secondary, contextually and culturally dependent element. This conjecture aligns with cultural narratives that often depict wise individuals, such as hermits or philosophers, who, despite their social detachment, are revered for their profound insights into virtuous living.”

It is interesting that people rated themselves better than others especially for socio-emotional awareness. The authors did not discuss what seems to me to be the more likely explanation that socio-emotional traits are less objective, and thus easier to view oneself positively. People tend to self-enhance more on subjectively defined traits. See Dunning, Meyerowitz, & Holzberg, 1989 JPSP.

Response: This is a great suggestion, we now add this explanation to the Discussion, P.13:

“In light of previous findings that self-assessments tend to be less accurate when evaluating desirable and behavioral characteristics (Vazire, 2010), and that they self-enhance on subjectively-defined traits (Dunning et al., 1989), our results suggest two potential explanations: first, people might value Socio-Emotional Awareness more than Reflective Orientation, leading to greater self-enhancement in this dimension; second, Socio-Emotional Awareness might have a more subjective nature, while Reflective Orientation might point to more directly observed characteristics and “objective” merits in others.”

The caveats that the authors listed here are important, and they do limit the generalizability of these findings. I think the authors should emphasize these limitations more in their discussion of their findings earlier in the paper. As it currently reads, the results sound far more conclusive until one gets to the caveats section.

Response: We revised the Discussion and used more careful phrasing to highlight the caveats and opportunities for future research.

Reviewer #2*Dimensions of Social Perception of Wisdom Across Five Continents**General thoughts:*

Conceptually, this is a really interesting paper and is very well written. It's particularly noteworthy that despite different epistemological traditions across cultures, people tend to agree on who is wise and which traits constitute wisdom. The authors should be applauded for their largescale cross-cultural approach to tackling the social perception of wisdom.

Response: Thank you so much!

This being said, I struggle with this paper's unique contribution to the social perception literature. It is somewhat unclear whether the two dimensions of wisdom perception differ meaningfully from dimensions of social perception identified in previous literature, nor whether these dimensions are uniquely capturing wisdom perceptions or some other related construct. Finally, the interaction effect between the two dimensions, while paradoxical and intriguing, merits further explanation and may actually challenge the authors' claim that wisdom perceptions vary along two dimensions.

Response: We provide an extensive revision to elaborate on the unique contribution of our work.

First, we highlight that our contribution is not chiefly to the social perception literature but rather to the related scholarship in cognitive science on *mind* perception.

Second, we highlight that both the mind perception and social judgment scholarship have a range of limitations concerning the existing psychometric methods applied to identify the underlying dimensions, as well as limited generalizability of the work from a cross-cultural perspective - the vast majority of social judgment research has been based on Western (European and North American) and a handful of East Asian samples. Moreover, whenever psychometric work was done, it was not done *across* cultures and whenever the broader set of cultures than North America or Europe were tested, psychometric testing was underdeveloped. Thus, our work is unique in its combination of robust state-of-the-art psychometric tests and the use of well-powered and diverse samples capturing societies from five different continents. Given that we started our project with the aim to examine mind perception with unique stimuli concerning concrete features of judgment in uncertain situations, it is quite remarkable that we identify dimensions that share some family resemblance with dimensions of social perception - a scholarship which relied on entirely different methods and target groups that had nothing to do with the topic of our inquiry.

We also elaborate on the nature and the meaning of the interaction. In preparation of our revision, we revisited our extensive Bayesian and frequentist analyses that revealed the interaction - our results hold across most cultural groups, outlining that at low and mid levels of reflective orientation, people who were rated as relatively higher in socio-emotional characteristics were attributed lower wisdom.

We elaborate on the meaning of this pattern - to perceive a person as wise, it is not enough to be *mindlessly* generous or emotionally sensitive. We don't think this result is controversial - perception of socio-emotional experiences may be important to qualify perception of one's reflective qualities. But all things being equal, one is more likely to perceive a person as wise, if they have a reflective mind rather than being a (mindless) slave of passion. Indeed, most of

us will know a person or two who are mindlessly driven by their feelings - our results suggest that such people would be attributed the lowest degree of wisdom.

These issues, along with several miscellaneous points, are outlined in greater detail below.

Response: We thank the reviewer for their thoughtful and clear suggestions; we have attempted to address each point below.

The Two Dimensions of Wisdom Perception

Core to the authors' claim is that perceptions of a target's wisdom vary along two dimensions: Reflective Orientation and Socio-Emotional Awareness. The authors acknowledge that these two dimensions are similar to previous work that has divided social perception along the dimensions of agency (or competence) vs experience (or warmth). But, looking at the items included in each respective dimension, it is unclear whether these new dimensions are distinct from the dimensions already described in the social perception literature. Reflective Orientation is comprised of control/logic/acting and Socio-Emotional Awareness is comprised of feeling/caring. The labels of the dimensions are of course just that, labels, and I think it's possible that the dimensions of wisdom perception are the same as the dimensions of social perception more broadly, in which case the question arises of whether this is revealing that universal dimensions of person/social perception (Susan Fiske's work) apply for judgments of wisdom, which are social perceptions. If warmth/feeling and competence/thinking are universal dimensions, then the authors might argue why it is novel to reveal them for judgments of wisdom.

Response: Overall, there is some resemblance between social perception dimensions from prior research and the wisdom perception dimensions revealed in our work. However, our project started from a very different angle (mind perception vs. social judgment) and applied very different methods (focus on psychological processes in the context of judgment vs. evaluation of trait-ascriptions to stereotyped groups). Thus, we view this very result as surprising. Moreover, future research might investigate the direct association between these two sets of dimensions, and we would expect that the association is not extremely strong. We stand with the novelty of our study for the following reasons:

- 1) Unlike SCM researchers, we aimed to address the question "How do people in different cultures think about wisdom?" Our initial idea was inspired by mind perception research - to explore how people compare mental characteristics among people we may associate with wisdom. We added additional comparison targets, as well as additional characteristics concerning emotion regulation, and sensory experiences, to expand the focus beyond the portrayal of wisdom-related characteristics in the West (per prior cross-cultural research and scholarship on beliefs about wisdom). The approach we took was very different from the typical studies on social perception (e.g., SCM/ABC) and social judgment, which typically uses adjectives (e.g., trusting, competent, warm) to evaluate stereotyped groups. From this perspective, it is remarkable that there is some convergence with the dimensions often discussed in social judgment research.
- 2) To elaborate on the difference between mind perception and social judgment, it is useful to remember that the focus of our research is the nature of good judgment in complex, uncertain situations, not how stereotyped groups are perceived. For our

research project, both Reflective Orientation and Socio-Emotional Awareness can be considered as dimensions reflecting unique, albeit inter-related forms of competence.

- 3) There is a very limited research of both mind perception - the inspiration for our work - and social perception in the Global South (with two exceptions that we acknowledge). Our scale, in contrast, includes eight cultural regions and 16 samples encompassing both the Global North and the understudied Global South.
- 4) Neither SCM nor mind perception research has directly compared the cross-cultural stability of underlying dimensions (i.e., they did not perform measurement invariance tests). Our study used state-of-the-art psychometric tests of measurement models and their cross-cultural extension to build on prior conceptual models in the scholarship on mind perception and social judgment.
- 5) Scholarship on dimensions of mind perception and social judgment has not been linked to the concept of wisdom (as well as concepts of knowledgeability and understanding). Our work explicitly tests how the latent dimensions of wisdom-related judgment relate to attribution of wisdom, knowledge, and understanding. Therefore, it provides novel information about criterion validity for the latent dimensions we identified.

Following these arguments, we rewrote the introduction and discussion to clearly show how our approach is different from the social perception studies, outlining how it builds and extends existing scholarship.

The Uniqueness of Wisdom Perception

One way to increase the novelty of these findings could be to show that the two dimensions pertaining to wisdom perception are uniquely associated with perceptions of wisdom, but they don't appear to be. Each dimension was positively correlated with explicit ratings of wisdom, as we would expect them to be, but the dimensions also showed similar correlations with explicit perceptions of knowledge and understanding. So again, are these dimensions uniquely capturing perceptions of wisdom, or some broader construct like intelligence or decision-making ability? If the authors are arguing this is special to wisdom, I think they need to do more to demonstrate that these dimensions are really capturing wisdom perception. Given the results, it feels like you might be able to replace "wisdom" with "knowledge" or "understanding" throughout the paper, even though each of these has different connotations. The fact that the two dimensions were not uniquely correlated with wisdom judgements could also be further evidence that they reflect the two universal dimensions of social perception?

Response: Despite the apparent similarity to the SCM dimensions (found a posteriori), we operate on a different analytical plane (evaluation of processes of judgment in an uncertain situation rather than abstract traits of stereotyped groups). As we outlined in an earlier response, we focused on evaluation of psychological characteristics in the context of making a judgment under uncertainty, with a focus on comparison of context-rich targets typically associated with wisdom (e.g., a scientist who gathers information about plants, animals, and people to make sense of the world). As we outline in the revision, both the psychological characteristics and the targets were selected from the prior wisdom scholarship. Thus, methodologically our research was firmly grounded in the quest for understanding the unique dimensions informing perception of wisdom across cultures.

We did not expect qualitatively different associations of the underlying dimensions with wisdom, knowledgeability, and understanding. As we wrote in the initial manuscript, we

treated attribution of wisdom as a criterion for evaluating the validity of the dimensions. Knowledge and understanding were introduced as “closely related epistemic attributes” (P.4) - i.e., these two terms were additional criterion validity probes to determine whether the underlying dimensions of wisdom perception would similarly impact related terms, beyond unique nuances of how these terms may be translated and understood across cultures. We did expect knowledge and understanding to produce similar (albeit not identical) results, particularly when it comes to knowledge – given the connection between wisdom and knowledge. For instance, in some languages such as Spanish the words translating wisdom and knowledge have the same lexical roots, and it is natural to think that people high on wisdom should be high on understanding. At the same time, the difference in connotations between wisdom and understanding might be clear in English, but it may not be the case in Chinese. Thus, using the terms wisdom along with knowledge and understanding would provide a strong test of criterion validity.

Puzzling Interaction Effect

The most interesting finding in the paper was the interaction effect between the two dimensions predicting wisdom, and this seems to be where the paper diverges from other social perception research. It's certainly puzzling that at the high end of reflectiveness, wiser people are those who have higher socio-emotional awareness, but that at the low end of reflectiveness, wise people have lower socio-emotional awareness. This leads to paradoxical patterns of results, like scientists being wiser than teachers despite having less socio-emotional awareness.

Given this finding, the two current dimensions seem to not fully capture how people think about wisdom. If someone of equal reflectiveness can have less socio-emotional awareness and yet be wiser, then there must be something else that is driving perceptions of wisdom. To make this more explicit, we can consider the case of mind perception, which is analogized in the manuscript. Mind perception varies along the two dimensions of agency and experience. Each dimension is positively associated with perceived mind: more agency  more mind at every level of experience, and more experience  more mind at every level of agency. But imagine that this research had instead found that having more agency sometimes leads to lower perceptions of mind. This would be a puzzle about perceived mind that demands an explanation.

Response: Following Reviewer 2's excellent suggestion to probe our results further, we performed further analyses to inspect the nature of the interactions and found that Reflective Orientation had a positive effect on wisdom at all levels of Socio-Emotional Awareness (see Fig. 3, also S8 and S9). On the other hand, effects of Socio-Emotional Awareness are negative at low and mid-levels of Reflective Orientation (see Fig. S7).

In the revision, we elaborate on these results, including a discussion of their meaning. Consider two people who are equally quite low on reflectiveness but one of them is more likely to be socio-emotional than another. This would be akin to a fool who is mindlessly (not reflectively) driven by emotions. A person who does not differentiate whom to help comes to mind. Or a person who passively follows their gut feeling without ever reflecting or showing signs of self-control. This tendency could manifest in indiscriminate helpfulness, akin to the well-established identifiable victim effect (e.g., Jenni & Loewenstein, 1997; Lee & Feeley, 2016, for a review) where decisions are driven by emotional responses to specific individuals rather than reasoned considerations of overall impact or broader ethical principles. Such

individuals might follow gut feelings without reflection or self-control, leading to decisions that, while prosocial, may not be prudent or wise in a broader sense.

To provide analogous examples from mind perception research: consider an agentic fool who is constantly “striving,” but their efforts chiefly reflect guesswork and very little experience. There are many such examples in real life (e.g., an overly eager student who is an ambitious go-getter, but naive and inexperienced would likely be perceived as more mindless compared to a similarly inexperienced student who is less eager/agentic). Thus, even though agency and experience are positively interrelated, there are cases where low scores on one dimension and higher scores on another dimension may be related to *lower* attribution of mind compared to comparably low scores on both dimensions.

To ensure greater clarity of these points, we also aim to elaborate on several core ideas.

What do the effects of the two dimensions on wisdom ratings indicate?

- 1) The Reflective Orientation is positively related to wisdom ratings but a little less so when SEA is lower suggesting that SEA is also necessary. We assume it is not surprising to expect both reflective and socio-emotional dimensions to achieve the highest wisdom ratings.
- 2) The main effect of SEA, when keeping Reflective Orientation constant, is negative (especially at low end of Reflection Orientation). This part requires some explanation (see below).
- 3) The interaction points to the fact that at the higher levels of Reflective Orientation, SEA does not have a negative association with wisdom and even suggests a positive association (see Fig. S7) – which again is not surprising and points to the fact that both dimensions are needed to achieve the highest wisdom ratings.
- 4) Interaction also points to the fact that at the lower ends of Reflective Orientation, SEA has a negative effect (as in the main effect of SEA, this and this part only is surprising).

What does the negative effect of SEA on wisdom ratings (at lower- and mid-level of Reflective Orientation) mean?

One important thing to keep in mind here is the *positive* zero-order correlation of SEA and wisdom ratings. The negative association between SEA and wisdom ratings is present only after holding Reflective Orientation constant.

This positive correlation between SEA and wisdom can be due to the well-established halo effect in social judgment (especially of desirable characteristics). We can control for this halo effect when examining partial correlations of each dimension with wisdom ratings (controlling for the other dimension). For instance, in the regression we can estimate each dimension’s unique contribution to wisdom, thus controlling for the halo effect. This analysis suggests that beyond the halo effect SEA has a negative association with wisdom ratings (especially at low levels of Reflective Orientation).

We now suggest the post-hoc interpretations of this result in the Discussion section.

1. Perception of wisdom positively depends on Reflective Orientation and negatively on Socio-Emotional Awareness - people are perceived less wise when they non-reflectively care about others. For instance, in our study the *scientist* who gathers information about plants, animals, and people to make sense of the world is considered wiser than the school *teacher* who educates twelve-year-olds about local history and literature despite the fact that the *teacher* cares more about others. By the same token, in our study the clearly caring *fair* person (an activist) is considered less wise than the *75-year-old* person whom participants considered relatively less caring. It appears that among the selected targets more caring is considered unwise, especially when the person is low on Reflective Orientation.
2. When a lower Reflective Orientation is combined with higher Socio-Emotional Awareness, it can lead to a stereotypical image of a less wise person (a striving fool), who demonstrates their lack of reflection clearly. Consider somebody who is all emotion and empathy but fails to show an ability to reflect, differentiate between various situations, and control oneself in the face of various situational demands. This person would likely be viewed as more foolish compared to an equally mindless person who is not driven by their passions. We give two accessible examples in the main text: people who give indiscriminately or people who give everything they possess.
3. Based on this interpretation, we can further speculate that Reflective Orientation is a necessary condition for being perceived as wise, and SEA helps with it only when the first condition is satisfied.

In addition, in the course of the robustness analyses, we ruled out the methodological explanation of the result – it is not an artifact caused by interaction at the higher levels of both dimensions, nor was it caused by the influential outlier (12-year-olds): exclusion of this target did not change the results. We provide details of these analyses in the revised supplement.

Miscellaneous Points

I think it would be useful to more explicitly define Reflective Orientation and Socio-Emotional Awareness in the intro, as opposed to just in the methods/results, since so much of the interpretation hinges on how we understand these dimensions. Maybe list a few more items that went into them, or just in simple terms how we might think of them.

Response: Since we derive these two dimensions from the results (the analysis was mostly exploratory), it wouldn't be practical to give these two dimensions a priori definitions. But we agree that it is a good idea to narrow down the definitions of these two dimensions. We expanded the interpretation part of the results to provide more insight into the meaning of the dimensions, writing in the revised introduction of these dimensions in the results section, P.6: "To interpret the meaning of each factor, we examined factor loadings (Fig. 2). We labeled the first factor *Reflective Orientation*, with high loadings of characteristics concerning thinking before acting, thinking logically and in many ways, recognition of change, emotion control, as well as application of knowledge and past experience. Overall, this dimension integrates pragmatic, rational, analytic, and self-control traits. This factor resembles some features discussed in prior research on mind perception and social judgment, which suggested "intentional agency"/"mind" (or "competence") as one dimension of judgment of mental states and groups.

We labeled the second factor *Socio-Emotional Awareness*, because of the highest loadings of characteristics concerning care for others' feelings, one's emotions, and for others' perspective, as well as humility (recognition that one might be wrong). This factor appears similar to the "conscious experience" that encompasses features of social meta-cognition, attention to the context and to one's emotions, as identified in some prior mind perception research (Gray et al., 2007). Notably, it combined both features of "heart"/socio-emotional characteristics and the "body"-related experiences (cf. Weisman et al., 2017; Willard & McNamara, 2019). Taken together, these characteristics describe traits concerned with social coordination and care for others."

We further provide details of features making up each dimension in the revised schematic Figure 1.

Similarly, I think it would be useful to provide a bit more detail in the intro about how people rated targets' wisdom. Were the pairwise comparisons dichotomous (e.g., scientist more logical than teacher) or continuous? Did the prompt ask participants to think about the targets in the context of a specific decision, or a difficult decision in general? These questions were answered later in the paper but were unclear the first time the measure was introduced.

Response: Thank you for highlighting this point. While we cannot provide the methodological details in the introduction (as per journal guidelines), we do so in the introductory sections of the opening of the results section when we introduce how people rated targets' wisdom (assuming this is what the Reviewer has in mind). We now write on P.4: "Participants compared ten human targets (including themselves) in a pairwise manner: They were asked whether one would be more likely than the other to act in a certain way when facing a difficult choice where there is no clear answer (e.g., "think logically," "care for others' feelings;" see Fig. 1 and Methods for further details). Each of the nineteen actions reflected a wisdom-related characteristic as discussed in prior research (Glück & Weststrate, 2022; Grossmann et al., 2020)."

In other words, the pairwise comparisons yield continuous (relative) scores, in a context of difficult decisions in general - the latter was done because we already had two types of variability (targets and actions) and hence it seemed prudent to keep the scenario type constant."

The word "dovetails" is used heavily – it might improve flow to vary the word choice, using terms like converges with/supports other research, etc.

Response: Agreed. We have minimized the use of this word in the revision.

In the first paragraph of the discussion, the phrase "our results suggest that the two latent dimensions of wisdom perception may be a psychological universal" feels like quite a strong claim. I feel like the authors can walk this back a bit and say that the effects were stable across cultures.

Response: Thank you for this excellent suggestion. In the revision, we state on P.12: "Overall, our results suggest that the structure of the two latent dimensions of wisdom perception is stable across very different cultures, although more work is needed in other

parts of the world to comprehensively test whether these two dimensions reflect psychological universals (Norenzayan & Heine, 2005).”

Concluding thoughts

In sum, I think the authors tackled an interesting question and I really appreciate the cross-cultural data and the elegant descriptive analyses. My uncertainty about the merits of the manuscript for Nature Comms revolves around the novelty. Just finding two dimensions—if they are the same universal dimensions of person/social perception—may not be sufficiently new especially if ratings of wisdom are the same as other constructs like knowledge. At the same time, there is clearly a wrinkle in wisdom ratings because of that interaction. That interaction might be the key to demonstrating novelty, but it is not sufficiently unpacked or explained in the current version.

Response: As we outline above (first response to Reviewer 2), we believe the methods, starting point of our research, and the sheer scope goes well beyond the conceptually and methodologically distinct scholarship on social judgment. As we elaborate in the expanded and revised introduction, prior research on mind perception and social judgment does not provide sufficient methodological tests of cross-cultural variability, nor does it formally test the measurement models (or formally probes their invariance) beyond North America, Europe and a handful of non-Western countries.

Furthermore, as we elaborate above, the results for ratings of knowledge (which shared an etymological root with wisdom in a number of languages such as Spanish - saber = “to know” and sabiduria = “wisdom”) and understanding (which pursues the same epistemic goals as wisdom) are additional probes of *criterion* validity of the dimensions of wisdom perceptions rather than tests of discriminant validity.

Finally, we hope that our expanded discussion of the interaction and the explanation of the observed pattern of results is not only compelling, but also demonstrates novel insights for the broad scholarship on wisdom, mind perception and perhaps even social judgment dimensions.

Another thing that might help is to think about the behavioral consequence of these dimensions. It seems like one reason that the warmth/competence work got so much traction is because it connected to behavior and other judgments (e.g., warmth is friend/foe). As the wisdom research line moves forward, I think it will be useful to consider not only whether Reflective Orientation and Socio-Emotional Awareness are unique from things like warmth and competence, but also whether each might have unique behavioral judgments. Are people more likely to trust the wisdom of certain dimensions in certain settings?

Response: The current research aimed at establishing dimensions of wisdom perception, and we see the study of convergent validity as a very important *next* step. It makes sense that these dimensions are useful to discern different types of competence needed for dealing with challenges and uncertain events in our lives.

At this point we can only speculate that trust might have an ambiguous association with both dimensions. Future studies could examine whether trust is related to Reflective Orientation when people are engaged in solving complex tasks, and to Socio-Emotional Awareness in interpersonal contexts (e.g., comforting someone grieving).

In the revision, we conclude by referring to this interesting line of future research, P.14:

“Moving beyond delineating wisdom perception dimensions, the next vital step is to assess whether the dimensions we have identified hold potential for differentiating key competencies in managing life's challenges. A particularly intriguing aspect for future research is whether people are more likely to trust individuals demonstrating unique features of wisdom in different contexts. For instance, are people more likely to trust individuals they perceive as high in Reflective Orientation in the context of complex problem-solving scenarios, while trusting individuals perceived as high on Socio-Emotional Awareness in the context of interpersonal dynamics, such as offering solace in times of sorrow. Investigating these relationships will deepen our understanding of wisdom's nuances and its diverse interpretations across cultures.”

Reviewer #3

This is a very interesting paper that aims to use an innovative methodology to identify a cross-cultural core of conceptions of wisdom. I applaud the authors for the immense effort invested in this study. At the same time, I have some concerns about several aspects of the methodology. I will try to explain these concerns in the following. As I am not familiar with some of the complex methods used for data analyses, I focus on the substantive meaning of the findings rather than on the technical decisions the authors made in analyzing the data.

Response: Thank you so much! We hope to have done justice to the concerns in the revised version.

First of all, the study involved ten different target characters: oneself, a 12-year-old, a 45-year-old, a 75-year-old, a religious person, a fair person, a teacher, a politician, a scientist, and a doctor. I am not sure I understand the rationale underlying this selection of characters. In the supplementary information, the authors say that “we selected a teacher, a scientist, and a doctor, because they are explicitly mentioned as exemplars of wisdom in prior North American literature (e.g., Weststrate et al., 2016). Finally, we selected a politician and a fair person (who is an activist fighting for human rights) as exemplars of civic leadership—another domain often associated with wisdom (Weststrate et al., 2016).” (page 3). So was the goal to select targets that participants would consider as wise? Then why not just use one target representing the wisest person participants know? In the Weststrate et al. study, some politicians (e.g., Lincoln, Churchill) were nominated as wise, but the target description might just as well make participants think of Donald Trump or their corrupt local mayor. Not every politician is considered as wise, nor is every teacher, doctor, or scientist.

Response: We did not aim to compile a list of equally wise targets; indeed, variability in degree of attributed wisdom to targets is something we would need for statistical analyses. Instead, our selection of targets was driven by the concerns that most of these targets should be theoretically associated with some form of wisdom in most societies, as well as by the cross-cultural translatability of targets. We included both wisdom exemplars and contrasting targets (e.g., 12-year-old). We opted to use several targets for two reasons. First, we expected that participants would assign varying levels of wisdom characteristics to different targets, and it was required to explore the underlying *within*-person dimensions (i.e., dimensions guiding one’s relative evaluation of pairs of targets on the psychological characteristics obtained from the same individuals) and distinguish them from the *between*-person dimensions (i.e., dimensions guiding individual differences in evaluation of target’s psychological characteristics; see Figure S2 in Supplementary Information for schematic differentiation between these levels of analysis). With just one target it would be a mixture of the two levels. Second, cultures were expected to differ in the degree of wisdom assigned to different characters. The idea of asking participants to pick somebody wise and then assess the characteristics of this participant-specific target is interesting and should be used in future research, but it can also jeopardize comparability of such ratings (e.g., some cultures may show preference for family members, the others for public figures, and then it would be impossible to infer whether we look at wisdom dimensions or family perception vs. public figure perceptions).

Also, participants in Japan and India were only presented with the targets “self,” “religious person,” and the three age groups, that is, with zero wisdom exemplars, so I’m not sure how much the data contributed to the final analyses.

Response: Using prior empirical research, “Religious person” and “75-year-old” were picked as exemplars of wisdom, self is neutral in this regard and 12- and 45-year-olds are non-wise exemplars, so the set of targets can be considered balanced yet of course there could be other possible selections of targets. We have added this explanation to the revised SI. Due to the multilevel structure of the models, the omitted targets were treated as ordinary missing values, and therefore did not have a power to change the results in a substantial way.

ML EFA on the pooled sample excluding Japan, and two Indian samples had only modest effect on the results. The optimal solution is still a two-factor solution and the structure of the factors is almost identical to the one reported for the full sample (see Table S11). Therefore, the data from the samples with a shortened list of targets did not have any substantial impact on the results. We now mention these new results in the SI.

Anyway, each participant was asked to compare one (randomly assigned) target to all of the other targets with respect to how likely they would show a total of 19 behaviors or ways of thinking in a difficult life situation. These characteristics were selected, according to the authors, because they are “associated with wisdom in prior philosophical and psychological scholarship on wisdom” (manuscript, p. 2). I am not sure which philosophical or psychological scholarship suggests that “maximizing the benefit for one’s group, regardless of the cost for others,” “not showing emotions,” or “paying attention to what nature tells you” is highly characteristic of wisdom.

Response: We have now extended the reasoning behind the item selection, please see the revised introduction on pp.2-3. We also revised the section about the role of prosocial elements, explicitly referring to the Common Wisdom Model and the role of moral grounding (incl. varied aspects of prosociality). Specifically, we write, P.17-18:

“We generated 19 characteristics following the core items from the previously established Common Wisdom Model (Grossmann et al., 2020; also see Grossmann & Kung, 2020), and similar frameworks featuring additional characteristics such as emotion regulation (Jeste et al., 2010). These items included meta-cognitive characteristics, as well as prosocial features such as cooperation. Further, we included two items referring to attention to one’s and others’ bodily expressions, based on the ideas that wisdom is associated with mindfulness (Jeste et al., 2010; Verhaeghen, 2020), and that attention and bodily awareness are central elements of mindfulness (Choi et al., 2021). This way, we sought to accommodate evidence from prior cross-cultural scholarship suggesting that in the non-WEIRD countries wisdom may be aligned with social, spiritual (or nature-related), and visceral experiences (e.g., Takahashi & Overton, 2002). Together, these characteristics corresponded to the dimensions of mind, heart, and body discussed in prior mind perception research (e.g., Weisman et al., 2017; 2022). To increase variance in the data, we also added one (reverse-coded) item concerning the lack of humility—showing pride in oneself, as well as one evolutionary adaptive, but arguably morally ambiguous feature concerning in-group favoritism. Each characteristic described a behavior, a mental action, or a focus of attention and did not explicitly refer to wisdom.”

Second, it seems that the characteristics are mostly representative of Western conceptions and those non-Western conceptions that have already been investigated. If the concept of wisdom in Japan or South America were very different from those in North America and Western Europe, how would this study detect that, given that neither the targets nor the characteristics would then be representative of wisdom in those parts of the world?

Response: We aimed at compiling a comprehensive list of items tapping into wisdom perception, including those typically assigned to the “West” and to the “East” (which though were not tested systematically across cultures), as well as some extra characteristics which were not yet investigated in relation to wisdom across cultures (e.g., bodily perception, pride).

We admit that the list might not be exhaustive. There was an operational limit to the list of items because otherwise the workload on participants would be too high. Furthermore, we had to field the same list of items in each sample to systematically investigate the underlying dimensions and test for their invariance.

We are sure there are some hidden and undiscovered facets of wisdom perception, as we acknowledge when we discuss the limitations of our project. Moreover, the new dimensions could appear had we studied other cultures (e.g., a huge variety of cultures in Sub-Saharan Africa and South-East Asia), as we now note toward the end of the paper.

We do not claim to have covered all possible dimensions. On the contrary, our claims were limited to the two dimensions which showed invariance across a given set of cultures.

Finally, we admit the limitations of the standardized approach in Discussion section, P.14: “Whereas the standardized format of our instrument to capture latent dimensions of wisdom perception allowed us to compare wisdom perception systematically across many societies, such questionnaire format may have fostered cross-cultural consistency in participants’ reports (e.g., Barrett, 2022). Future research might explore whether employing natural-language processing methods to analyze open-ended narratives about wise individuals yields similar cross-cultural consistencies.”

Participants provided their comparisons on five-point scales ranging from “much less likely” to “much more likely” – e.g., they would judge whether their target, the doctor, was less or more likely to think about the issue in many different ways than a 12-year-old. While this method provides an impressive number of data points, I did not really understand what its advantages are in comparison to simply asking participants to rate the different targets on the different characteristics. Which potential biases or distortions are eliminated by using pairwise comparisons instead, especially when the comparisons are between different figures that are all assumed to be (relatively) wise? In other words, why does information on how participants in different cultures describe a doctor in comparison to, say, a religious person tell us more about their conceptions about wisdom than just looking at how they describe a doctor (or perhaps simply how they describe a very wise person)? It would be great if the authors could explain that in the paper.

Response: Not all targets were assumed to be wise. To ensure we have sufficient variance for psychometric analyses, we aimed at creating a list of targets with varying degree of wisdom (and including some a priori relatively low wisdom targets such as a 12-year-old as a sanity check).

The pairwise comparison method was initially planned to be used with a matching multidimensional scaling technique, following past studies on mind perception by Gray and

colleagues (2007). Other than that, the choice of method was meant to control for response bias prevalent in surveys with rating-scale questions. Moreover, the response style tendencies for rating scales differ across cultures and therefore can bias the results of the cross-cultural comparisons (e.g., He & Van De Vijver, 2015). Now we clarified this in the Methods -> Procedure section.

After completing a rather long list of comparisons, participants were also asked to rate all 19 targets with respect to wisdom, understanding, and knowledgeability (in addition to the translation difficulties with the word “understanding” that the authors mention, it would also be interesting to know how they ensured the comparability of the words for “wisdom” in the different languages).

Response: First, we relied on our extensive team of experts in cross-cultural research, linguistics, and anthropology, who were heavily involved in translating, adopting and vetting all materials. We have extensive discussions about the terms to use in each culture, and specifically the terms wisdom, but also knowledge and understanding. We also relied on classic back-translation techniques and expert agreements about the terms to use across sites. The revised version explicates it in the Methods -> Procedure section.

Second, to further test robustness of results across different terms reflecting epistemic goals associated with wisdom, we also examined patterns of results for knowledge and understanding, yielding comparable results for attribution of knowledge, and similar (albeit slightly different in ways that are not relevant to the current point) results for understanding.

Third, in the course of additional analysis we also constructed a latent variable out of the three variables (wisdom, knowledge, understanding) and showed partial metric invariance across cultural regions. Factor loadings for all three items appeared to be similar across cultural regions – which points to similarity in meaning of these terms across eight cultural regions, with exception of the ‘understanding’ item at the within-person level. Indeed, understanding has also slightly deviated in its associations with the two perception dimensions. We report these new results in the SI.

To clarify these methodological details in the revised manuscript, we now write at P.17:

“Firstly, we enlisted an extensive team of experts in cross-cultural research, linguistics, and anthropology. These specialists played a crucial role in the translation, adaptation, and validation of all materials. Secondly, in-depth discussions were conducted focusing on key terms like wisdom, knowledge, and understanding, ensuring cultural appropriateness and semantic accuracy. This process involved classical back-translation techniques and consensus-building among experts to finalize the terms used across different cultural sites. Additionally, research teams at each site were encouraged to report any challenges encountered during the translation of philosophical terms, facilitating team deliberations to resolve ambiguities and align interpretations in the target language.”

Analyses were highly complex because of the complex data structure, but they showed that quite universally, the 19 characteristics grouped into two factors. This does not seem particularly surprising given that the authors themselves say that other research has already shown the cross-cultural existence of two cardinal dimensions of social judgment. Looking back at the target characters listed above, it makes a lot of sense that many of them (scientist, doctor, teacher) would be considered as high in the Reflective Orientation but not necessarily

the Socio-Emotional Awareness factor. That might also explain why participants in most regions considered themselves as higher than the targets in Socio-Emotional Awareness but not in Reflective Orientation.

Response: The existing empirical literature, which we now present at greater length in the introduction, does not speak unequivocally for a two-factor model. Rather, it is consistent with one-, two-, or even three-factor models, and we regarded these models as equally plausible at the beginning of our project.

Furthermore, as we now make it clear in the introduction, the existing literature doesn't provide sufficient evidence of the universality of social judgement (e.g., Stereotype Content Model) nor mind perception (see comments above), and trends in the literature that we now review in more detail would lead us to expect variation in the dimensions underlying wisdom perception. In fact, we expected variation, and we were genuinely surprised by our finding of an invariant two-factor structure.

We also did not know the dimensions of wisdom perception in advance: it *is* surprising that the perception of wisdom-related characteristics and the social perception of groups would converge on comparable, though not necessarily identical, dimensions, particularly when one keeps in mind the methodological differences between research on social perception and our project (which we now emphasize in the article and have discussed in the remainder of this letter)

It is also not obvious that, e.g., the teacher target (which was, more precisely, described as follows: “Dr. [name] is a school teacher who educates twelve-year-olds about local history and literature.”) is very higher than *doctor* on Socio-Emotional Awareness and lower on Reflection Orientation, and it's definitely surprising that teacher's Reflective Orientation and Socio-Emotional Awareness are similar across different cultures.

In addition, we offer a new interpretation of the self-evaluations (see ref to Dunning et al., 1989 above). Specifically, in the revision we write P.13:

“In light of previous findings that self-assessments tend to be less accurate when evaluating desirable and behavioral characteristics (Vazire, 2010), and that they self-enhance on subjectively-defined traits (Dunning et al., 1989), our results suggest two potential explanations: first, people might value Socio-Emotional Awareness more than Reflective Orientation, leading to greater self-enhancement in this dimension; second, Socio-Emotional Awareness might have a more subjective nature, while Reflective Orientation might point to more directly observed characteristics and “objective” merits in others. Investigating these possibilities will allow us to refine our understanding of wisdom perception and how individuals may be biased in their assessments.”

The strongest evidence for the 19 characteristics indeed being representative of wisdom comes from the correlations between participants' wisdom ratings of the various targets and their judgments of the 19 characteristics. If participants rated a target as wiser, they also rated that target as more likely to show the behaviors on the Reflective Orientation factor and, to a lower extent, the Socio-Emotional Awareness factor. Basically, it seems to me that these findings show that participants rated, say, a doctor as wiser than a 45-year-old (see Figure S7) and that they also considered the doctor as higher in at least the Reflective Orientation factor. I'm not sure that the findings provide strong evidence that the

characteristics represented by the two factors are the key characteristics that people across the world consider as typical for wisdom. There seems to be some circularity in the fact that the researchers selected the characteristics from existing conceptions of wisdom, then showed that they are indeed correlated (one of them not even strongly) with ratings of wisdom.

Response: We agree that there could be *other* characteristics related to wisdom and we do not claim to present an exhaustive set of all possible wisdom dimensions (see the discussion of the limitations of the study). Yet we are confident that the two dimensions we have identified do – to some degree – underlie wisdom perception. Here is why:

When one performs psychological assessment of convergent validity, the standard operating procedure in psychology is to create a new instrument and use existing instruments (here, explicit ratings of wise persons) to establish convergent validity of the new instrument. At least since Campbell & Meehl's classic treatment of construct validity, this is considered evidence that the instrument measures what it's supposed to measure. In reference to Campbell and Meehl's seminal work on construct validity, our approach aligns with the established standard that an instrument is considered valid if it measures what it purports to measure. Here, we acknowledge the reviewer's concern regarding the potential circularity inherent in this standard validation method within psychology. To mitigate this concern, we emphasize the importance of using criterion variables that are external and distinct from the indicators of the new instrument. By employing criterion variables that are sufficiently distant from the instrument's own indicators, we reduce the risk of circular reasoning. This approach ensures that the validation process is not merely self-referential but is anchored in external, objective measures, thereby strengthening the argument for the instrument's construct validity.

To be concrete, our study followed these construct validation considerations with extra caution. First, none of the items in our pool indicated wisdom, knowledge, or understanding *directly*. Instead, the items were descriptive and more specific than broad terms like “wise.” These items could form two consistent dimensions, *or not*, which in turn could be indicative of the *explicit* wisdom judgments across different cultures, *or not*. To our surprise only a few items deviated from the two-dimensional structure of wisdom perception. In the revised manuscript we now emphasize these points and highlight the unexpected nature of the measurement invariance (see Research Overview, p.4; Attribution of wisdom and related epistemic content section of the Results, p10).

More generally, from a philosophical and methodological point of view, the development of any measure involves some circularity (see historian and philosopher of science Hasok Chang's (2004) classic work on the development of thermometry since the 17th century. The relevant question is whether the circularity is *vicious*: that is, whether the results are guaranteed. As we have just argued, there were many opportunities for our study to produce very different results.

I do think there is some highly interesting evidence in this data set, but some of it may be in the differences between regions rather than in the commonalities.

Response: We appreciate the reviewer's insight into the potential value of regional differences within our data set. However, it is crucial to note that our study relies on convenience samples, which inherently limit our ability to draw definitive conclusions about cultural differences. Such samples may reflect idiosyncratic characteristics tied to the specific

composition of the groups rather than broader cultural trends. Therefore, while regional differences in the data are indeed intriguing, they may not necessarily signify true cultural variances.

On the other hand, the similarities we observed across different regions in the dimensional structures of our instrument are of particular significance. The similarity of dimensional structures is less likely to appear by chance and is even less likely to appear due to differences in sample composition. The consistent patterns across diverse regions suggest a robustness in the instrument's measurements, underscoring its potential utility across different cultural contexts.

Acknowledging Reviewer 3's perspective, we have expanded sections of the manuscript to more comprehensively describe the differences observed between samples from various regions. However, as outlined in specific responses below, we maintain a cautious interpretation of these differences, emphasizing that they should not be conclusively linked to cultural disparities alone due to the limitations of our sampling method. This approach balances the intriguing nature of regional variations with the methodological constraints of our study, ensuring a nuanced and accurate representation of our findings.

- *For example, I found it very interesting that nature and divinity had higher loadings on the Socio-Emotional Awareness factor in traditional communities.*

Response: It is indeed very interesting to look at the differences in factor loadings. We carefully examined the difference in factor loadings across cultural regions (Fig. S3), but did not spot any systematic difference except the general suggestion we mentioned in the revised manuscript. Specifically, we state on P.8:

“Though speculative, cross-site variability in the value of “nature and divinity” for the Socio-Emotional Awareness dimension may reflect stronger socio-cultural emphasis on nature and divinity in indigenous communities in Indian (Meitei) and South African (isiZulu and Sepedi) samples—the outliers in this item’s loadings on the Socio-Emotional Awareness dimension.”

Furthermore, we write on P.10:

“Effect of Reflective Orientation was significant and positive in all eight regions for ratings of wisdom, knowledgeability, and understanding (except for non-significant effect on understanding in China). The effects of Socio-Emotional Awareness on wisdom ratings were non-significant in North America, South America, and Morocco, but it was negative in the other five regions. When examining attribution of knowledgeability, the effects of Socio-Emotional Awareness were negative and significant in all regions but Morocco. Finally, Socio-Emotional Awareness had negative effects on attribution of understanding in India and South Africa, but positive in all the other regions except South America and Morocco where it was non-significant.”

- *Also, did the wisdom ratings of the different targets vary between regions?*

Response: We agree it is an interesting question, but because it's not central to our paper, it was only briefly mentioned in regard to Fig. S7 where one can find a representation of variability of ratings across regions for the explicit ratings of wisdom, knowledgeability, and understanding. In addition, we now extended a report on it in the SI:

“The average ratings of wisdom, knowledgeability, and understanding in each cultural region were highly stable. Across regions, an average intercorrelation of mean ratings of wisdom was $r = .91$, for knowledgeability $r = .94$, and it was a little lower for understanding, $r = .80$.”

- *For example, did participants everywhere consider the religious person as equally unwise? Did such variations perhaps go along with differences in characteristics ascribed to the targets?*

Response: Thank you for your insightful question regarding the perception of wisdom in religious figures across different cultural samples. To address this, we must first clarify the scope and methodology of our analysis:

- a. Our study primarily conducts relative, within-person comparisons within each sample, focusing on how participants rank different targets against each other. This within-person analysis allows us to understand how individuals evaluate various figures relative to each other, as well as in terms of relative judgments who is wiser. It is not designed to make absolute comparisons between different targets across diverse samples (which is the question of between-person differences; see our earlier response to the question about differences between these levels of analysis).
- b. Regarding your specific inquiry about the average ratings of wisdom attributed to religious figures across different cultures, our approach is cautious. The reliance on convenience samples in our research introduces potential biases that might reflect the specifics of each sample rather than genuine cultural differences. Consequently, comparing mean ratings of wisdom for each target across samples may not yield reliable insights into cultural variances. Instead, it might inadvertently highlight idiosyncrasies within each sample group.

Given these methodological considerations, we have chosen not to report on the differences in mean ratings of wisdom for each target across all samples. We believe that this decision helps maintain the integrity of our findings, emphasizing the structural comparisons within samples rather than potentially misleading cross-cultural comparisons. It's crucial to recognize that while such nuances and specific questions are intriguing, they may extend beyond the reliable interpretative scope of our study's design and data.

- *In addition, there seem to be considerable differences in sample compositions across regions, especially with respect to education, which should be documented in more detail in the supplemental materials.*

Response: Thank you for this excellent suggestion! We added a *Sample composition section* to SI, expanding on sample composition across regions.

- *For example, why were there so many male participants in the Slovak sample?*

Response: This is one of the downsides of using convenience samples – in Slovakia the survey was run in university classes, and the classes happened to consist of primarily male students. We should note that (a) our analyses matched the gender of targets to the gender of participants to avoid some gender-specific effects; (b) we controlled for both participant's and target's gender in our analyses when examining cultural differences and other patterns of

results; (c) we also observed some interesting secondary results pertinent to effects of target's gender. Specifically, as we write in the revised manuscript P.18:

“We used this two-level structure to fit exploratory and confirmatory factor analyses, and then extended it to the multilevel structural equation models to test latent dimensions' associations with explicit attributions of wisdom. The latter controlled for individual differences including age, gender, education, and religiosity.

“Furthermore, we write, P.11: “Researchers from each cultural site picked the gender of the targets deemed culturally appropriate. Therefore, we controlled for the target's gender when examining differences in ratings between targets. Furthermore, we tested how targets' gender is associated with wisdom perception. The results showed that female targets were rated lower than male targets on Reflective Orientation, albeit comparable to male targets on Socio-Emotional Awareness. This result reminds one of the prior research on social judgment and gender stereotypes (White & Gardner, 2009), expanding it beyond the WEIRD samples used in most prior scholarship.”

- *To what extent do differences in participant age, gender, or education account for some of the regional differences? (I noticed that the Slovak sample had the lowest correlation between Socio-Emotional Awareness and wisdom ratings, and Table S23 seems to suggest that this may be a gender effect.)*

Response: This is a good point and we admit that convenience samples do not allow making a clear inference about differences in cultures – as noted above.

I understand that it is probably difficult to look for differences in this rich trove of data without clear theory-based hypotheses, so there may be other ways to enhance the incremental value of this study for our knowledge about cultural differences in wisdom – right now, I am not quite sure how much we can really learn from it.

Response: In the revised Discussion, we expanded on some points of group differences: 1) on differences in rankings across cultural regions, 2) on differences in associations between the two dimensions and explicit wisdom ratings across cultural regions; 3) on the gender differences we observed in our analyses. Specifically, female targets were rated lower than male targets on Reflective Orientation, albeit comparable to male targets on Socio-Emotional Awareness.

Small issues:

Page 3: “Contrary to our expectations, these dimensions were consistent across cultures:” what were the authors' expectations and why?

Response: We have now clarified our prior expectations in the introduction. Prior research on mind perception and social judgment suggested either a one-dimensional solution, a two-dimensional solution, or even a three-dimensional solution. Based on much prior anthropological and cultural psychological research, we further expected differences across cultures. Based on prior wisdom work and some general claims about "collectivism," it is conceivable that in East and South Asia a socio-emotional dimension would be central to the ascription of wisdom, while a reflection dimension would be relatively less important, while the opposite would be the case in the West. Based on the same literature, we also expected a

reflection dimension to predict the explicit wisdom judgments in the West, and the socio-emotional dimension in East and South Asia.

Page 8: “in real life people typically evaluate others based on limited information about them.” I do not quite see that providing one sentence of description increases ecological validity – in comparison to what?

Response: Our point is that we provided more context to each target compared to a typical setup in the social judgment literature (which uses, e.g. a single adjective or a label like “the rich” or “elderly”). In the revision, we elaborate on this point and the usefulness of some additional context. Now the text goes, P.13-14:

“This approach was deliberately chosen to mirror the complexities of real-life social evaluations, where individuals often form judgments based on limited information, and the attributes being compared are seldom mutually exclusive. By adding some context about each target, as well as providing sentence-long descriptions of psychological characteristics, we further expanded our methodology beyond the typical settings in existing research on social and mind perception, where subjects are often described abstractly and limited to one or two words. Our strategy aimed to enhance the ecological validity of our study by introducing relatively more nuanced and context-rich scenarios.”

Reviewer #4

Based on the statistical review of the manuscript, it seems that the authors do an excellent job explaining the multi-level analysis results to the domain expert. The explanations of the factor analysis results, particularly, were very clear and precise. It is evident that the authors have a good understanding of this concept. The figures, especially factor loadings (Figure 2), were also found to help clarify the analysis. Furthermore, the code used for the analysis was found to be precise and reproducible.

Response: Thank you so much!

However, the reviewer recommends that the authors clarify in the text that a Bayesian approach was used. Although it is clear from the code that Bayesian analysis was used, nothing in the text suggests it. It would be beneficial to comment on what prior was used for which parameters and provide an explanation suggesting Bayesian analysis. Overall, the statistical analysis of this manuscript was found to be correct.

Response: We fully agree and now elaborate on this point in the revised Methods section. Please also note that we used Bayes only for prediction of the factor scores, whereas (for the sake of consistency) fit a maximum likelihood estimator for all the other models.

References

- Barrett, L. F. Context reconsidered: Complex signal ensembles, relational meaning, and population thinking in psychological science. *American Psychologist* **77**, 894–920 (2022).
- Chang, H. (2004). *Inventing temperature: Measurement and scientific progress*. Oxford University Press.
- Dunning, D., Meyerowitz, J. A. & Holzberg, A. D. Ambiguity and self-evaluation: The role of idiosyncratic trait definitions in self-serving assessments of ability. *Journal of personality and social psychology* **57**, 1082 (1989).
- Epley, N. & Waytz, A. Mind perception. *Handbook of social psychology* **1**, 498–541 (2010).
- Glück, J. & Weststrate, N. M. The wisdom researchers and the elephant: An integrative model of wise behavior. *Personality and Social Psychology Review* **26**, 342–374 (2022).
- Grossmann, I. *et al.* The science of wisdom in a polarized world: Knowns and unknowns. *Psychological Inquiry* **31**, 103–133 (2020).
- He, J. & Van De Vijver, F. J. Effects of a general response style on cross-cultural comparisons: Evidence from the Teaching and Learning International Survey. *Public Opinion Quarterly* **79**, 267–290 (2015).
- Jenni, K., & Loewenstein, G. (1997). Explaining the identifiable victim effect. *Journal of Risk and uncertainty*, *14*, 235-257.;
- Jeste, D. V. *et al.* Expert consensus on characteristics of wisdom: A Delphi method study. *The gerontologist* **50**, 668–680 (2010).
- Lee, S., & Feeley, T. H. (2016). The identifiable victim effect: A meta-analytic review. *Social Influence*, *11*(3), 199-215.
- Norenzayan, A. & Heine, S. J. Psychological universals: What are they and how can we know? *Psychological bulletin* **131**, 763 (2005).
- Peng K., Nisbett R., Wong N. Validity problems of cross-cultural value comparison and possible solutions. *Psychological Methods* **2**, 329–341 (1997).
- Takahashi, M. & Overton, W. F. Wisdom: A culturally inclusive developmental perspective. *International Journal of Behavioral Development* **26**, 269–277 (2002). Tesser A., Shaffer D. (1990). Attitudes and attitude change. *Annual Review of Psychology*, *41*, 479–523.
- Vazire, S. Who knows what about a person? The self–other knowledge asymmetry (SOKA) model. *Journal of personality and social psychology* **98**, 281 (2010).
- Verhaeghen, P. The examined life is wise living: The relationship between mindfulness, wisdom, and the moral foundations. *Journal of Adult Development* **27**, 305–322 (2020).
- Weisman, K. *et al.* Similarities and differences in concepts of mental life among adults and children in five cultures. *Nature Human Behaviour* **5**, 1358–1368 (2021).
- Weisman, K., Dweck, C. S. & Markman, E. M. Rethinking people’s conceptions of mental life. *Proceedings of the National Academy of Sciences* **114**, 11374–11379 (2017).
- White, J. B. & Gardner, W. L. Think women, think warm: Stereotype content activation in women with a salient gender identity, using a modified Stroop task. *Sex roles* **60**, 247–260 (2009).
- Willard, A. K. & McNamara, R. A. The minds of god (s) and humans: Differences in mind perception in Fiji and North America. *Cognitive science* **43**, e12703 (2019).

Reviewers' Comments:

Reviewer #1:

Remarks to the Author:

I was a reviewer for the previous draft of this manuscript. I thought it was very strong then, and I think the authors have done an excellent job in responding to the reviewers' comments. I have no further suggestions for revisions. I think this is an excellent paper.

Reviewer #2:

Remarks to the Author:

The authors did an excellent job addressing my initial questions. I especially appreciate the effort they took in explaining the puzzling negative interaction between the two dimensions. I buy their explanation about how it seems unwise to be driven so much by emotions.

I also appreciate the stronger theoretical connection to the dimensions of mind perception, because the potential parallels and divergences are worth outlining.

Finally, I laughed when I saw just how much content the authors managed to cram into the two-sentence summary.

Reviewer #3:

Remarks to the Author:

I very much appreciate the authors' thoughtful considerations of the points made by all reviewers (as well as the great effort that went into this study). At the same time, my main concerns about this study have not really been alleviated. To be clear, I think this is a very interesting study, but the insights it produces are more incremental than truly substantial. In responding to my comments, the authors say that some aspects of the data, such as differences between national samples in mean levels of wisdom ascribed to different targets, are not valid because the samples all were convenience samples (and, as I mentioned in my review, differ considerably in demographic characteristics). Thus, the main finding of the study is that when participants compare a set of pre-defined target persons on a set of different characteristics, the factor structure underlying these comparisons (which is very nicely shown in Figure 4) is pretty stable across cultures. This factor structure essentially distinguishes a cognitive and a socio-emotional dimension, because the items included essentially describe a cognitive and a socio-emotional dimension. As I mentioned in my first review, it would be very interesting to see whether people in all the different cultures equally associate the individual items with wisdom (I would assume that there are interesting differences there), but such differences may not manifest themselves in how much they associate the individual items with teachers or 45-year-olds. In sum, I still have doubts about the importance of the contribution of this paper to our knowledge about cultural commonalities and differences in conceptions of wisdom.

Reviewer #4:

Remarks to the Author:

The authors have satisfactorily addressed the concerns raised.

Response to Reviewers

We are very thankful to all the reviewers for their appreciation of our update of the manuscript.

REVIEWERS' COMMENTS	
Reviewer #1 (Remarks to the Author):	
I was a reviewer for the previous draft of this manuscript. I thought it was very strong then, and I think the authors have done an excellent job in responding to the reviewers' comments. I have no further suggestions for revisions. I think this is an excellent paper.	Thank you!
Reviewer #2 (Remarks to the Author):	
The authors did an excellent job addressing my initial questions. I especially appreciate the effort they took in explaining the puzzling negative interaction between the two dimensions. I buy their explanation about how it seems unwise to be driven so much by emotions. I also appreciate the stronger theoretical connection to the dimensions of mind perception, because the potential parallels and divergences are worth outlining. Finally, I laughed when I saw just how much content the authors managed to cram into the two-sentence summary.	Thank you!
Reviewer #3 (Remarks to the Author):	
I very much appreciate the authors' thoughtful considerations of the points made by all reviewers (as well as the great effort that went into this study). At the same time, my main concerns about this study have not really been alleviated. To be clear, I think this is a very interesting study, but the insights it produces are more incremental than truly substantial. In responding to my comments, the authors say that some aspects of the data, such as differences between national samples in mean levels of wisdom ascribed to different targets, are not valid because the samples all were convenience samples (and, as I mentioned in my review, differ considerably in demographic characteristics). Thus, the main finding of the study is that when participants compare a set of pre-defined target persons on a set of different characteristics, the factor structure underlying these comparisons (which is very nicely shown in Figure 4) is pretty stable across cultures. This factor structure essentially distinguishes a cognitive and a socio-emotional dimension, because the items included essentially	Thank you!

describe a cognitive and a socio-emotional dimension.	
As I mentioned in my first review, it would be very interesting to see whether people in all the different cultures equally associate the individual items with wisdom (I would assume that there are interesting differences there), but such differences may not manifest themselves in how much they associate the individual items with teachers or 45-year-olds. In sum, I still have doubts about the importance of the contribution of this paper to our knowledge about cultural commonalities and differences in conceptions of wisdom.	We thank Reviewer for their encouragement to consider item-wise analyses!. We now report them in the main text and the supplement: Parallel analyses with raw items (instead of factors) revealed that all sixteen items were positively correlated with wisdom ratings across the six regions with only minor exceptions (see Table S28). Further regression analyses with all items as simultaneous predictors of wisdom ratings revealed that in each region the largest significant effects came from ‘thinking logically, thinking in many ways’, ‘applying experiences,’ and ‘control of emotions’—all part of the Reflective Orientation dimension (see Table S29 in SI). Regarding Reviewer’s concern about the list of targets that we used (assuming it is their main concern), we selected them following the literature review and discussions across the study sites: These targets are the most frequently nominated wise exemplars, along with two control targets, 45-year-old and 12-year-old. Our revised manuscript explicitly acknowledged this methodological detail. In other words, 45-year-old and 12-year-old are not exemplars of wisdom. Our manuscript also acknowledges that further empirical evidence is needed to expand this research beyond the present list of targets.
Reviewer #4 (Remarks to the Author):	
The authors have satisfactorily addressed the concerns raised.	Thank you!
Reviewer #4 (Remarks on code availability):	
The code is very clear and reproducible.	Thank you!